# The Gastrointestinal Exertional Heat Stroke Paradigm: Pathophysiology, Assessment, Severity, Aetiology and Nutritional Countermeasures

**DOI:** 10.3390/nu12020537

**Published:** 2020-02-19

**Authors:** Henry B. Ogden, Robert B. Child, Joanne L. Fallowfield, Simon K. Delves, Caroline S. Westwood, Joseph D. Layden

**Affiliations:** 1Faculty of Sport, Health and Wellbeing, Plymouth MARJON University, Derriford Rd, Plymouth PL6 8BH, UK; cwestwood@marjon.ac.uk (C.S.W.); jlayden@marjon.ac.uk (J.D.L.); 2School of Chemical Engineering, University of Birmingham, Birmingham B15 2QU, UK; robchild@elitesportgroup.org; 3Institute of Naval Medicine, Alverstoke PO12 2DW, UK; Joanne.Fallowfield258@mod.gov.uk (J.L.F.); simon.delves216@mod.gov.uk (S.K.D.)

**Keywords:** exercise, sport, nutrition, supplement, gut

## Abstract

Exertional heat stroke (EHS) is a life-threatening medical condition involving thermoregulatory failure and is the most severe condition along a continuum of heat-related illnesses. Current EHS policy guidance principally advocates a thermoregulatory management approach, despite growing recognition that gastrointestinal (GI) microbial translocation contributes to disease pathophysiology. Contemporary research has focused to understand the relevance of GI barrier integrity and strategies to maintain it during periods of exertional-heat stress. GI barrier integrity can be assessed non-invasively using a variety of in vivo techniques, including active inert mixed-weight molecular probe recovery tests and passive biomarkers indicative of GI structural integrity loss or microbial translocation. Strenuous exercise is strongly characterised to disrupt GI barrier integrity, and aspects of this response correlate with the corresponding magnitude of thermal strain. The aetiology of GI barrier integrity loss following exertional-heat stress is poorly understood, though may directly relate to localised hyperthermia, splanchnic hypoperfusion-mediated ischemic injury, and neuroendocrine-immune alterations. Nutritional countermeasures to maintain GI barrier integrity following exertional-heat stress provide a promising approach to mitigate EHS. The focus of this review is to evaluate: (1) the GI paradigm of exertional heat stroke; (2) techniques to assess GI barrier integrity; (3) typical GI barrier integrity responses to exertional-heat stress; (4) the aetiology of GI barrier integrity loss following exertional-heat stress; and (5) nutritional countermeasures to maintain GI barrier integrity in response to exertional-heat stress.

## 1. Introduction

Exertional heat stroke (EHS) is a life-threatening medical condition involving total thermoregulatory failure, which is the most severe condition along a continuum of heat-related illnesses [1]. Although anecdotal records have documented mortality from EHS as far back as biblical times [2,3], to the present day, EHS still has no universal medical definition [4]. Instead, the most popular definitions broadly outline characteristic patient symptoms at time of clinical admission [5]. These principally include: (1) a core body temperature (T_core_) above 40 °C; (2) severe central nervous system disturbance (e.g., delirium, seizures, coma); and (3) multiple organ injury. Whilst classic heat stroke (CHS) primarily impacts incapacitated individuals (e.g., elderly, infants, chronic illness) whose thermoregulatory responses are insufficient to compensate against increased ambient temperatures [6], EHS sporadically impacts individuals (e.g., athletes, military personnel, firefighters) engaged in arduous physical activity [7]. Indeed, the primary cause of EHS is extreme or prolonged metabolic heat production, whilst exposure to high ambient temperature is less important than in CHS cases, despite further compromising thermoregulation [8].

The incidence of EHS has been frequently surveyed within high-risk populations since the beginning of the 20th century [3]. Despite this, issues surrounding misdiagnosis (e.g., with less severe heat illness events) have generally limited accurate classification [9,10]. Over the last two decades, the annual incidence of EHS has remained relatively stable in both athletic [11] and military [12] settings. Indeed, prevailing EHS incidence rates are reported to be *circa*: 0.1–1.5 cases per 10,000 US high school athletes per season [13,14]; 0.5–20 cases per 10,000 entrants during warm weather (ambient temperature ≥ 25 °C) endurance races [15,16,17]; and 2–8 cases per 10,000 person years in both the United Kingdom [18] and United States [12,19] armed forces. Given global predications of increased ambient surface temperature, coupled with a greater frequency, duration and intensity of extreme weather events, the risk of EHS is only anticipated to increase in the future [20]. Timely medical intervention (e.g., whole-body cooling within 1 h of EHS symptom onset) offers nearly a 100% chance of survival from EHS [21], though many affected individuals still experience long-term health complications because of residual organ damage. These health-complications include: heat intolerance [22], neurological impairment [23], chronic kidney disease [24] and cardiovascular disease [25]. The burden of EHS not only relates to the health of the patient, but can also result in reduced occupational effectiveness [26,27], significant medical/legal expenses [28,29], and in some instances high-profile media criticism [30,31] of concerned governance bodies (e.g. employer). In consideration of these issues, numerous consensus documents have been published that provide occupational guidance on effective EHS management (e.g., [32,33,34,35]), though predominately take a thermoregulatory approach to disease management (e.g., cooling, heat acclimation). A gastrointestinal paradigm of EHS pathophysiology (also known as “endotoxemia” or “heat sepsis”) has gained momentum as a secondary pathway through which to focus EHS management [36,37], though consensus documents detailing this approach are currently unavailable.

The gastrointestinal (GI) tract, is an organ extending the stomach to the colon. It is the human body’s longest mucosal interface (250–400 m^2^), which forms a selectively permeable barrier between the GI lumen and circulating blood. The GI microbiota is a collection of microorganisms that colonise the GI tract, which have co-evolved inside humans and are considered to provide several mutually beneficial host functions [38]. The GI microbiota has an estimated size circa 10^14^ cells, which is estimated 1- to 10-fold greater than the total number of cells of the human body [39]. Alongside a predominant role in the absorption of dietary nutrients, a second vital function of the GI tract is to prevent the translocation of immunomodulatory GI microbial products (e.g., endotoxin, flagellin, bacterial DNA) into the systemic circulation [40]. To achieve this role, the structure of the GI tract is comprised of a multi-layered physical and immunological barrier. The physical barrier comprises a monolayer of epithelial cells interconnected by tight junction (TJ) protein complexes, and is reinforced by a mucosal lining secreted by goblet cells. The immunological barrier comprises crypt paneth cells within the epithelial monolayer that secrete antimicrobial proteins, and gut associated lymphoid tissue within the lamina propria that stimulate multiple effector immune responses. In healthy individuals, the GI tract is largely effective in preventing GI microbial translocation (MT) into the systemic circulation [40], however, growing evidence hypothesises a fundamental role of GI MT within the pathophysiology of EHS [36,37]. The focus of this review is to evaluate: (1) the GI paradigm of EHS; (2) GI barrier integrity assessment techniques; (3) typical GI barrier integrity responses to exertional-heat stress; (4) the aetiology of GI barrier integrity loss; and (5) nutritional countermeasures to support GI barrier integrity during exertional-heat stress.

## 2. The GI Exertional Heat Stroke Paradigm

The GI EHS paradigm was first introduced as a novel pathophysiological concept in the early 1990s [41], before integration into conventional EHS medical classifications in 2002 [5]. The broad scientific basis of the GI EHS paradigm centers the notion that sustained exertional-heat strain initiates damage to the GI barrier, which consequently permits GI MT into the circulating blood. To counter this response, the liver’s reticuloendothelial system (RES) provides the first line of GI microbial detoxification (e.g., Kupffer cells and hepatocytes) through the portal circulation. However, the RES only confers a limited capacity for microbial neuralization before leakage into the systemic circulation arises [42]. Alternatively, GI MT can bypass the RES altogether, instead translocating directly through the mesenteric lymph nodes into the systemic circulation [43]. In the systemic circulation, MT products are neutralized through multiple host-binding pathways, which include: natural antibodies (e.g., immunoglobin G and M), leukocyte granular proteins (e.g., bactericidal permeability increasing protein, lactoferrin, lysozyme) and high-density lipoproteins [42]. In EHS patients, microbial detoxification capabilities are likely reduced, predominately due to the combined effects of RES dysfunction at T_core_ above ~41–42 °C [44] and immune antibody suppression following strenuous exercise [45]. Failure of GI microbial detoxification mechanisms permits binding of unique GI pathogen associated-molecular patterns (PAMP) to toll-like receptors (TLR) located on cell surface membranes [46]. TLR activation initiates a cascade of intracellular events (e.g., nuclear factor kappa-light-chain-enhancer of activated B cells) that initiate the production of pro-inflammatory cytokines (e.g., interleukin [IL] 1-β, IL-2, IL-6, IL-8, tumor-necrosis factor [TNF]-α), which are counter regulated by the production of anti-inflammatory cytokines (e.g., IL-1ra, IL-4, sIL-6r, IL-10, sTNFr). Downstream of this systemic inflammatory response syndrome (SIRS), a complex interplay of responses that potentially culminates in hemorrhagic shock, disseminated intravascular coagulation (DIC), multiple organ failure (MOF) and possibly death [47]. The GI EHS paradigm is considered to be the primary cause of EHS in cases where T_core_ remains below the threshold (~42–44 °C) of heat cytotoxicity [48]. A simplified schematic of the GI EHS paradigm is shown in Figure 1. Interested readers are referred to several detailed reviews on this topic [1,36,37].

To date, direct investigation into the pathophysiology of the GI EHS paradigm has been limited, which is perhaps surprising given the substantial morbidity/mortality associated with the disease. Constrained by ethical restrictions to study EHS in healthy humans, best available evidence is unfortunately reliant on experimental animal models of CHS/EHS or opportunistic monitoring of human EHS patients in clinical field settings. In a pioneering study, canines who were administered antibiotics prior to CHS (peak T_core_ = ~43.5 °C) were found to exhibit both lower GI microbial stool concentrations and an improved survival rate (71% versus 20%) compared with control animals [49]. Although this study did not directly assess GI MT, the authors hypothesized this to be a primary mechanism [49]. In a series of seminal studies, adopting a primate CHS model (peak T_core_ = ~43.5 °C), plasma endotoxin concentrations increased parallel to T_core_ during passive heating (50–52), however, prior treatment with either antibiotics [50,51] or corticosteroids [52] attenuated this response. Importantly, across these studies 100% of prior-treated animals survived, in comparison with less than 30% of control animals. However, once hyperthermia was above the intensity to evoke heat cytotoxicity (peak T_core_ = ~44.5 °C), mortality rates were 100% irrespective of pharmaceutical intervention. Together, these findings suggest the GI EHS paradigm to be most relevant in cases when T_core_ remains below ~42–43 °C [48]. In recent years, several studies have confirmed these findings in comparable rodent models of CHS (peak T_core_ = ~43.5 °C). For example, prior corticosteroid injection was shown to inhibit GI MT and improve survival [53,54,55], whereas indomethacin injection enhanced gross morphological GI hemorrhage and worsened survival [56]. Similarly, direct intravenous endotoxin injection prior to sub-lethal CHS in rodents (peak T_core_ = ~42–43 °C) unexpectedly led to fatalities in 40% of animals (versus 0% in controls; [57]), or best-case increased symptoms of multiple-organ injury [58]. At present, only one study has assessed the role of GI barrier integrity utilising a relevant EHS model, whereby rats run to thermal collapse in the heat (T_core_ = 40.5–42.5 °C) displayed significant histopathological damage to all GI segments [59,60], together with a significant pro-inflammatory response [61]. However, in comparison to CHS models with a similar clinical endpoint (peak T_core_ = ~42–42.5 °C), the magnitude of GI barrier integrity loss was lower following EHS, though this finding is potentially attributable to a ~50% lower thermal area [60]. To date, no published animal research has attempted to evaluate the role of GI MT on EHS pathophysiology. Inconsistent with EHS GI paradigm, recent data demonstrate the pattern of cytokine response during EHS to be largely inconsistent to those displayed following GI microbial PAMP recognition (e.g., minimal TNF-α/IL-1β response; [60]. With this is mind, it is plausible that cytokine production initiated in response multiple organ injury (e.g., skeletal muscle; [61]) performs a greater role in EHS pathophysiology than previously proposed [37,47].

In humans, the role of GI barrier integrity within EHS pathophysiology is a poorly characterised research area, developed off the back of historical reports of severe GI symptoms, ulceration and hemorrhage in military EHS fatalities [62,63,64]. Direct evidence supporting the GI EHS paradigm was first reported by Graber et al. [65], who observed systemic endotoxin translocation and symptomology of septic shock in a single EHS case report. More substantial evidence was collated in the 1990s from EHS patients (peak T_core_ = ~42 °C) presenting at clinical settings during religious pilgrimage to Mecca [66]. The plasma endotoxin concentration of these patients was ~1000-fold greater than in healthy controls (8.6 ng·mL^−1^ vs. 9 pg·mL^−1^). In this study, weak correlations were reported between endotoxin and SIRS responses (e.g., TNF-α *r* = 0.46; IL-1β *r* = 0.47), whilst in a follow-up study that did not monitor endotoxin responses, IL-6 concentration weakly correlated (*r* = 0.52) with the disease Simplified Acute Physiology Score (SAPS; [67]). Similarly, in a cohort of military EHS patients (peak T_core_ = ~41.5 °C), IL-2 (*r* = 0.56), IL-6 (*r* = 0.57) and IFN-γ (*r* = 0.63) concentrations weakly correlated with the SAPS, though no associations were evident between the time-course of any other cytokine monitored (IL-1β, IL-2ra IL-4, IL-8, IL-10; TNF-α) [68]. Finally, IL-6 and sTNFR, but not IL-1ra or C Reactive Protein, predicted survival in a later cohort of EHS patients (peak T_core_ = ~41.5 °C) on Mecca pilgrimage [69]. Whilst none of these studies directly monitored GI MT responses, sub-clinical exertional-heat stress (T_core_ = < 40 °C) experiments report similar patterns of endotoxin translocation and SIRS kinetics in some [70,71], but not all cases [72,73]. A key limitation of previous human research into the GI EHS paradigm has been the exclusive reliance of endotoxin as a biomarker of GI MT. To this end, there is substantial evidence that blood samples may become cross contaminated during sample collection or analysis. For example, in one EHS case study, the presence of β-glucan (a fungal cell wall component) in the blood was shown to be the main determinant of the initial positive endotoxin reading [74]. One novel biomarker that might offer better sensitivity/specificity in diagnosing GI MT is procalcitonin (PCT), a pro-inflammatory acute phase reactant that is commonly endorsed for confirming bacterial infection during sepsis [75]. In EHS patients, PCT measured 2 h following intensive care unit admission was able to predict Acute Physiology and Chronic Health Evaluation (APACHE) II score (*r* = 0.59) and had an odds-ratio of 2.98 for predicting disease mortality [76]. Furthermore, in CHS patients, PCT concentrations were significantly greater in fatal versus non-fatal cases [77,78].

## 3. Assessment of GI Barrier Integrity

Various techniques are available for the in vivo assessment of GI barrier integrity. These techniques can be broadly categorised as either: (1) active tests involving the oral ingestion and extracellular recovery of water-soluble non-metabolizable inert molecular probes; (2) passive tests involving monitoring blood biomarkers indicative of GI barrier integrity; and (3) microbial translocation (MT) tests involving monitoring blood biomarkers indicative of the passage of GI microbial products across the GI barrier secondary to integrity loss (Table 1 [40]).

The Dual Sugar Absorption Test (DSAT) is presently promoted as the gold-standard active GI function test [79] and has received almost exclusive application with the field of exercise science [80,81,82]. This test involves co-ingestion of both a large disaccharide (e.g., lactulose [342 kDa] or cellobiose [342 kDa] ~5 g) that only transverses the GI tract paracellularly upon barrier integrity loss, and a small monosaccharide (_l_-rhamnose [164 kDa] or _D_-mannitol [182 kDa] ~1–2 g) that freely transverses the GI tract transcellularly independent of barrier integrity [83]. In the five hour period post-ingestion, the excretion of both sugars are measured in urine and are believed to be equally affected by non-mucosal factors, such as gastric emptying and renal clearance [84]. The urinary ratio of lactulose-to-rhamnose (L/R) relative to the ingested dose is the clinical endpoint of this test. Recently, the DSAT has been validated in serum/plasma with improved sensitivity over a time-courses ranging between 60 and 150 min [85,86,87,88], and with comparable reliability to traditional urinary assessment [89]. Unfortunately, the DSAT has several practical limitations, most notably: a requirement to perform basal/exercise tests on separate days and a lack of universal test standardisation (e.g., pre-trial controls, sugar dose, ingestion timing, biofluid timing) [84]. Furthermore, based on the degradation of lactulose in the large intestine, the test only provides information regarding small GI barrier function, with further sugar probes (i.e., multi-sugar absorption test; MSAT) required to assess gastroduodenal (e.g., sucrose/rhamnose; S/R) and large intestinal (e.g., sucralose/erythritol; S/E) barrier function [82]. Whilst routine implementation of the MSAT would be desirable, hyperosmolar stress utilising recommended sugar dosages will cofound the test result. In an attempt to overcome this issue, validation of a low dose (1 g lactulose, sucrose, sucralose; 0.5 g _l_-rhamnose, erythritol) MSAT protocol has recently been favorable evaluated against the traditional dose (5 g lactulose, 2 g _l_-rhamnose) DSAT protocol [87,90]. Polyethylene glycols (PEG; 100–4000 kDa) are a less-common, though a validated alternative to the MSAT for whole-GI barrier integrity assessment [91]. An advantage of PEG assessment is the ability to provide information on the size based permeability of molecules able to transverse the GI barrier. However, this method does require additional lifestyle controls, as PEGs can be found in various commercial/dietary products (e.g., toothpaste, soft drinks) [82]. The application of single molecular probes tests (e.g., non-metabolizable sugars, ^51^Cr-EDTA, Iohexol, Blue #1 Dye) cannot be recommended in exercise settings given the confounding influence of non-mucosal factors [84].

Several passive blood-based biomarkers of GI barrier integrity are available, which can assess epithelial injury to specific regions of GI tract, TJ breakdown and MT [40]. Epithelial injury to the duodenum and jejunum can be evaluated via intestinal fatty-acid binding protein (I-FABP); and to the ilium via ileal bile-acid binding protein (I-BABP). These cytosolic proteins are involved in lipid metabolism, though offer strong diagnostic specificity/sensitivity in detecting GI barrier integrity loss [92], given their tissue specificity and transient 11 min half-life [93]. Alternative biomarkers of GI epithelial/transmural injury include: alpha-glutathione s-transferase (α-GST), diamine oxidase (DAO) and smooth muscle protein 22 (SM22); however, a lack of tissue specificity limits their application in settings (e.g., exercise) where multiple-organ injury is commonplace [40,94]. There is presently no available biomarker of large intestinal epithelial injury. To assess TJ breakdown, zonulin, a pre-curser protein to haptoglobin, has received most widespread attention, given its recognised role in disassembling GI TJs [95]. However, the two commercial assays presently available for this biomarker are susceptible to cross-reactivity (e.g., for complement protein C3). Consequently, data collected with this technique should be interpreted with caution until the methods have been validated [96]. Claudin-3, is a non-tissue specific, highly expressed GI TJ protein, which is an emerging biomarker for TJ breakdown. Preliminary data have shown that claudin-3 concentrations are elevated in clinical conditions where GI TJ damage has been confirmed histologically [97]. The test–retest reliability of I-FABP and claudin-3 was recently considered acceptable when assessed both at rest and following exertional-heat stress [89]. All GI epithelial injury/TJ breakdown biomarkers can be assayed in plasma/serum by ELISA, whilst future developments in auto-analyser’s and validation of capillary blood and urine samples have potential to make assessment simpler in the future.

The definition of MT was traditionally founded on the translocation of live bacteria from the GI lumen into the mesenteric lymph. However, given practical constraints of mesenteric lymph biopsy in healthy humans, this definition has been extended to include the detection of microbial products/fragments in blood [98]. To determine GI MT, measurement of endotoxin, a form of lipopolysaccharide (LPS) located on the outer membrane of Gram-negative bacteria, has been widespread [80]. Endotoxin is detectable within the portal/systemic circulations following bacterial cleavage during both cell lysis and division, with assessment widely undertaken using the chromogenic limulus amoebocyte lysate (LAL) assay. Whilst popular, there are major flaws to endotoxin assessment, as it is prone to false-positive (e.g., from exogenous contamination, cross-reactivity) and false-negative (e.g., from hepatic clearance, immune neutralization) results [99]. Two indirect surrogate biomarkers for endotoxin exposure that can be quantified by ELISA are the acute phase proteins: lipopolysaccharide binding-protein (LBP; [100]) and soluble-CD14 (sCD14-ST; [100]). Whilst the roles of these biomarkers have been characterised during life-threatening septic shock [101], evidence regarding their time-course, sensitivity and specificity in predicting transient GI MT following exertional-heat stress is sparse [80]. D-lactate is a secondary enantiomer of L-lactate, hypothesised as a biomarker of GI MT given that the enzyme D-lactate dehydrogenase is specific to bacteria [102]. That said, human cells do produce small quantities of D-lactate through secondary methylglyoxal metabolism [102]. Whilst D-lactate has been shown to predict GI MT in animal models of gut trauma [103,104], its low-molecular weight (0.09 kDa) might permit false-positive results through transcellular translocation following production within the GI tract. Bacterial DNA (bactDNA) is a stable bacterial component, which through targeting phyla with high GI specificity offers potential as an improved MT biomarker [105]. Whilst a universal analytical procedure is currently lacking (e.g., target primers, positive/negative controls), one major advantage of bactDNA over endotoxin assessment, is an apparent lack of rapid hepatic clearance [46]. As the GI microbiota is dominated (≥90%) by two bacterial phlya *Firmicutes* and *Bacteroidetes*, which comprise only a minor proportion (0–10%) of the whole blood/plasma microbiota [106], developing methodologies that target these specific gene regions are likely to provide high GI specificity. Pioneering studies have shown total 16S DNA to offer good reliability at rest and post exertional-heat stress, however *Bacteroides* DNA (the dominant *Bacteroidetes* bacterial genus) offered poor reliability at both time points [89].

## 4. Severity of GI Barrier Integrity Loss Following Exertional-Heat Stress

Numerous research models have characterised the influence of exertional-heat stress on GI barrier integrity. This research has primarily monitored small intestinal integrity using the DSAT, though attempts have been made to quantify gastroduodenal and large intestinal integrity using the MSAT [80]. Over the last decade, several passive GI integrity and/or MT biomarkers have become commonplace as an alternative to, or for use in combination with the DSAT. Generally, I-FABP has been monitored to assess GI epithelial integrity, and endotoxin to assess GI MT. The exercise models assessed are disparate, ranging from 45 min brisk walking [107] to a multiday ultramarathon [71]. That said, most studies comprise 1–2 h of continuous, submaximal (60–70% VO_2max_) running or cycling. Given the hypothesised relevance of GI barrier integrity within the pathophysiology of EHS, the impact of exercise-induced thermal strain (e.g., T_core_) on GI barrier integrity has been a specific topic of investigation [81]. In comparison to acute exercise interventions, few studies have attempted to evaluate the effect of either chronic exercise training or multi-day occupational performance (e.g., sports competition, military/firefighting operation) on GI barrier integrity. Such exercise models would appear particularly relevant to EHS incidence, given that many documented EHS risk factors (e.g., prior heat exposure, skeletal muscle injury) relate to multi-day exercise [37]. Review tables are provided to summarise the effects of acute exercise on: DSAT (Table 2); I-FABP (Table 3); and MT (Table 4).

Seminal research using the DSAT investigated the effects of one hour’s treadmill running in temperate conditions on GI barrier integrity [108]. These authors found that the DSAT ratio increased relative to both the magnitude of metabolic (60%, 80% and 100% VO_2max_) and thermal (38.0, 38.7 and 39.6 °C T_core_ peak) strain [108]. Later studies monitoring GI barrier integrity following exercise in temperate conditions corroborated this seminal finding, with low-to-moderate intensity (~40–60% VO_2max_) exercise having little influence on DSAT results compared with rest (e.g., [109,110,111]); whereas moderate-to-high intensity (~70–120% VO_2max_) exercise of durations ≥20 min increase permeability by 100–250% (e.g., [86,88,112,113,114,115,116]). Unfortunately, the present data do not allow more specific conclusions to be drawn, given large intra-study variability in absolute DSAT ratios, which can be attributed to modifications in the DSAT procedure (e.g., sugar probe type/dose/timing, analytical protocol) and/or a frequent lack of basal GI permeability correction (Table 2). That said, individual studies highlight the importance of particular aspects of the exercise stimulus on GI barrier integrity, with increased DSAT ratios after matched interventions comparing: running and cycling [112]; permissive dehydration versus rehydration [117,118]; and following ingestion of non-steroidal anti-inflammatory drugs (NSAID) [119,120,121,122,123]. To date, only two published studies have directly compared the influence of ambient temperature on GI barrier permeability [115,124]. In conflict with a priori hypotheses, the first of these studies found two hours of moderate intensity (60% VO_2max_) treadmill running in temperate (22 °C/44% relative humidity [RH]) versus mild hyperthermic (30 °C/35% RH) conditions resulted in comparable DSAT responses (0.025 ± 0.010 vs. 0.026 ± 0.008 [124]). However, these results were perhaps not entirely surprising given that T_core_ responses showed minimal divergence between the two environmental conditions (e.g., peak T_core_ = 38.1 °C vs. 38.4 °C [124]). A follow-up trial on the same subjects compared the results of the temperate exercise condition (22 °C/44% RH) with a third trial conduced in a more severe hyperthermic (35 °C/26% RH) environment [115]. The DSAT data (0.032 ± 0.010) remained statistically indifferent to the temperate condition, despite greater T_core_ elevations (e.g., peak T_core_ = 39.6 °C [115]). These null findings might be interpreted with caution, as there was poor analytical reproducibility of sugar concentrations (duplicate sample coefficient of variation = 13.8%) and no basal DSAT correction.

In comparison with the extensive literature examining the acute effect of exercise on small GI integrity using the DSAT, few studies have assessed the influence of exercise or exertional-heat stress on either gastroduodenal or large GI barrier integrity utilising the MSAT [80]. In the only published evidence where the MSAT was applied with reference probe co-administration [82], both gastroduodenal (S/R; [123]) and large intestinal (S/E; [86]) integrity were unaltered following one hour of moderate intensity cycling (70% watt_max_) in temperate conditions (~22 °C), which was sufficiently intense to induce detectable small intestinal barrier integrity loss using the DSAT. Similarly, gastroduodenal integrity, measured using a single sugar-probe (sucrose) has been shown to be unaltered following one hour of moderate intensity treadmill running (40–80% VO_2max_) in temperate conditions [108,118,121], 18 repeated 400 metre supramaximal track sprints (120% VO_2max_) in temperate conditions [88] and a ~33 min exercise capacity trial at 80% ventilatory threshold in the heat (35 °C/40% RH [125]). No further studies have measured large intestinal integrity following acute exercise using a single sugar-probe (sucralose). There is a clear gap in the literature regarding the influence of exertional-heat stress on large intestinal integrity, which warrants future investigation given the greater microbiota concentration in this segment of the GI tract (e.g., duodenum = <10^3^, ilium 10^3^–10^7^, colon = 10^12^–10^14^) [38].

Application of I-FABP as a biomarker of epithelial injury in the duodenum and jejunum was first used in exercise settings in a series of studies conducted in the Netherlands, which found concentrations to peak (~50–100% increase) immediately following termination of a one-hour moderate-intensity (70% Watt_max_) cycle [86,123,126]. I-FABP responses showed weak correlations with I-BABP (i.e., ileum injury) and the DSAT [86], suggestive of inconsistent injury across the small intestine. Since then, low intensity exercise (~50% VO_2max_) in temperate environments has typically shown little effect on I-FABP concentrations [137,138,139], but moderate-to-high intensity exercise (60–120% VO_2max_) elevates concentrations by 50–250% [88,124,140,141,142]. Where measured, I-FABP responses quickly recover within 1–2 h of exercise termination, irrespective of the intensity/duration of the protocol [124,140]. Like DSAT results, I-FABP responses are elevated in otherwise matched exercise interventions comparing: hypoxic (F_i_O_2_ = 0.14) versus normoxic environments [137,143]; permissive dehydration versus rehydration [144]; and post NSAID ingestion [123]. In comparison, since initial investigation [86], no studies have monitored the magnitude and time-course of I-BABP responses following exercise. Several studies have attempted to elucidate the influence of ambient temperature on GI epithelial injury [115,124,142,145,146]. Compared with modest increases in I-FABP (127%) following two hours of moderate intensity cycling (60% VO_2max_) in temperate (22 °C/44% RH) conditions (peak T_core_ 38.1 °C), performance of matched exercise in both mild (30 °C/35% RH [115]) and severe heat stress conditions (35 °C/26% RH; [124]) vastly enhanced peak T_core_ (38.4 °C and 39.6 °C) and percentage change in I-FABP (184% and 432%) responses, respectively. Furthermore, a moderate correlation (*r* = 0.63) was shown between peak T_core_ and I-FABP concentration in these studies. Ingestion of cold (7 °C) relative to temperate (22 °C) water during two hours moderate intensity cycling (60% VO_2max_) in the heat, blunted the rise in both T_core_ (38.4 vs. 38.8 °C) and I-FABP (~400% vs. 500%) concentration [146], though whether these responses are directly related is questionable. These conclusions were recently substantiated following one hour of low intensity (50–70% watt_max_) cycling, where I-FABP concentration increased following performance in a hot (35 °C/53% RH; 140%), but not temperate (20 °C/55% RH; 29%) ambient environment [142]. Importantly, these observations have been directly attributed to the influence of ambient temperature on whole-body thermal strain, given that when relative exercise-intensity is matched (VO_2max_, T_core_, heart rate), the influence of ambient heat stress (20 vs. 30C°) on I-FABP responses is abolished [145]. One study reported GI TJ breakdown (claudin-3) to increase to a similar extent following one hour of running in a temperate (22 °C/62% RH) versus hot (33 °C/50% RH) ambient environment [147], suggestive that TJ breakdown is insensitive to thermal stress. Alternatively, I-FABP and claudin-3 responses positively correlated (*r* = 0.41) following an 80-min brisk walk (6 km·h^−1^/7% incline) in the heat (35 °C/30% RH) [89].

Endotoxin is a traditionally popular biomarker of GI MT and was the first technique utilised to assess GI barrier integrity in exercise settings. Seminal research monitoring endotoxin concentrations following exercise, found concentrations to increase transiently to magnitudes comparable to clinical sepsis patients (~50–500 pg·mL^−1^) when measured following competitive ultra-endurance events [80]. These included: an ultra-triathlon [159], a 90 km ultra-marathon [160], a 100-mile cycle race [161] and a 42.2 km marathon [162]. More recently, only minor increases in endotoxin concentrations have been shown following comparable duration competitive ultra-endurance races [71,163,164], whilst moderate intensity exercise (≤2 h; 50–70% VO_2max_) performed in a temperate environment generally does not influence circulating endotoxin concentrations [141,142,147,162]. These discrepant results may be due to cross-contamination from β-glucan during early research, which following development of more robust endotoxin assays is now less of an issue [163]. It appears a presently undefined threshold of GI barrier integrity loss is required to induce endotoxemia following exercise, given that endotoxin concentrations are often unchanged from rest irrespective despite concurrent rises in DSAT or I-FABP concentrations [116,124,141]. When endotoxin is assessed from systemic blood samples, hepatic/immune detoxification might lead to false-negative results, and in exercise settings access to portal blood is rarely feasible. Given the large range in absolute endotoxin concentrations reported between studies (Table 4), several recent attempts have been made to measure MT with alternative biomarkers, though results are equally inconsistent [134,140,165,166]. Thermal stress appears to enhance endotoxin translocation above matched exercise performed in temperate conditions. In an early study, endotoxin concentrations increased linearly above 38.5 °C when (measured at 0.5 °C T_core_ increments), during uncompensable (40 °C/30% RH) treadmill walking (4 km·h^−1^) [165]. Likewise, a follow-up study found one hour of moderate intensity treadmill running (70% VO_2max_) only increased endotoxin concentrations in hot (33 °C/50% RH; 54%), but not temperate (22 °C/62% RH) conditions [147]. In a series of studies monitoring endotoxin concentrations following two hours moderate intensity treadmill running (60% VO_2max_), concentrations were found to increase by 4–10 pg·mL^−1^ irrespective of the thermal environment (22–35 °C) [115,124,167]. Numerous other studies have measured endotoxin concentrations following exertional-heat stress, though large intra-study variability in absolute concentration make it impossible to make precise recommendations regarding the typical magnitude of response (Table 4). In studies where endotoxin concentrations do increase following exertional-heat stress, responses peak immediately upon trial termination [147,168].

Whilst many studies have monitored GI barrier integrity responses following acute exertional-heat stress, relatively few studies have monitored GI barrier integrity following chronic (multi-day) exertional-heat stress. Where chronic exercise studies have been undertaken, they predominately focus on the influence of structured heat acclimation on GI barrier integrity. In an early study, involving seven days fixed-intensity heat acclimation (100 min walking at 6.3 km·h^−1^ in 46.5 °C/20% RH), endotoxin concentrations remained stable both at rest and following exertional-heat stress, despite T_core_ peak above 39.0 °C [162]. Utilising a variation of this experimental design, five consecutive days treadmill running at lactate threshold pace in the heat (40 °C/40% RH) until T_core_ had risen 2 °C above rest, evoked comparable post-exercise I-FABP and endotoxin responses compared to day-one [72]. Likewise, 10 days of fixed-intensity heat acclimation (one hour running at 50% VO_2max_ in 40 °C/25% RH), had no influence on post-exercise I-FABP concentration compared to day one [137]. In a recent study, neither seven nor thirteen days isothermic heat-acclimation (90 min to sustain T_core_ ~38.5 °C) blunted the rise in endotoxin concentration following 45 min low intensity (40% watt_max_) cycling in the heat (40 °C/50% RH), despite large reductions in thermal strain [173]. In a non-heat acclimation study, 14 days of 20% increased training versus standard load, led to a reduction in resting endotoxin concentration (35%), but did not influence peak concentrations following a 70% VO_2max_ treadmill run (35 °C/40% RH) until a T_core_ of 39.5 °C was attained [173]. The influence of aerobic fitness has been shown to both increase (I-FABP; [152]) and reduce (endotoxin; [165]) GI barrier integrity loss following exertional heat stress that evoked comparable thermal strain between groups. Future research, using well-designed and adequately powered studies coupled with sensitive biomarkers, is required to determine the influence of heat acclimation on GI barrier integrity. As well as ensuring an appropriate sample size, an exertional-heat stress protocol that evokes high physiological strain should be used, using study participants that posse the same physiological characteristics as the target population.

## 5. Aetiology of GI Barrier Integrity Loss following Exertional-Heat Stress

The aetiology of exertional-heat stroke induced GI barrier loss appears multifactorial and is incompletely understood. The best supported explanations relate to: hyperthermia-mediated dysregulation of GI TJs [176]; splanchnic hypoperfusion-mediated ischemia-reperfusion injury [82,177]; and alternations in several complex neuroendocrine-immune related interactions [178].

Increased tissue metabolic rate during strenuous exercise, and/or environmental heat stress, can evoke uncompensable heat strain on the body as thermoregulatory cooling responses (e.g., sweating and increased skin perfusion) become overwhelmed [179]. Within the GI tract, exertional-heat stress results in a relatively uniform rise in tissue temperature across both the small and large intestinal segments (though this rise is lower in the stomach), which can be predicted from T_core_ assessment in the distal colon [180]. This will weaken the GI barrier by morphologically disrupting the enterocyte structure and opening TJ complexes [176]. Cell culture models have consistently shown temperature elevations from 1.3 °C to rapidly disrupt the GI barrier in a dose/duration-dependent manner [181]. Rodent studies support these conclusions, with evidence of both histopathological GI damage and increased GI permeability following passive heating >40 °C [182]. Nevertheless, the mechanistic pathways directly linking hyperthermia to GI barrier integrity loss have been poorly characterised. The available evidence suggests that heat stress positively regulates the GI barrier through sodium-dependent glucose cotransporter/tyrosine kinase pathways [183] and negatively through the myosin light-chain kinase/protein kinase-c pathways [184]. Ethical constraints have prevented laboratory GI barrier integrity assessment following severe hyperthermia (>40 °C) in humans. However, a systematic review including available data up until September 2016 reported strong correlations (*r* = 0.91) between peak T_core_ and GI barrier integrity loss when all available (T_core_ and GI barrier integrity) assessment techniques were included [81]. Data presented in Table 2, Table 3 and Table 4 show a weak correlation between peak post-exercise T_core_ (rectal, gastrointestinal or oesophageal) with peak I-FABP (Δ; *r* = 0.52; *p* = <0.001), but not the DSAT (5-h urine only; *r* = 0.30; *p* = 0.19), or endotoxin (Δ; *r* = 0.14; *p* = 0.56) concentration (note: studies without T_core_ assessment were excluded).

Splanchnic vascular beds receive ~20% of total resting cardiac output but consume only 10–20% of the available oxygen [185]. Consequently, blood flow during strenuous exercise can be safely redistributed from splanchnic organs to skeletal muscle to maintain aerobic metabolism, and to skin to assist thermoregulation [179]. Hypoperfusion of splanchnic vascular beds, measured using doppler ultrasonography, appears to be proportional to exercise intensity and duration [185]. Specifically, splanchnic blood flow declines by 30–60% following both 30 min of moderate-intensity (60–70% VO_2max_) and 1–2 h of low-intensity exercise (40–50% VO_2max_) [186]. These responses appear amplified when exercise is performed in a warm environment [187]. A key downstream event following GI hypoperfusion is GI ischemia measured using gastric tonometry, which is also known to be suppressed following exercise in an intensity-dependent manner [86,188]. Localised GI hypoperfusion is considered to evoke secondary adenosine triphosphate depletion, acidosis, altered membrane ion pump activity and oxidative stress, all physiological responses that damage the GI barrier [181,182,189]. One limitation of this research is the inability of tonometry to measure large intestinal ischemia in exercising humans, especially as the largest microbial biomass is located in the distal GI segments [190]. The partial pressure of oxygen across the GI tract displays a proximal-to-distance gradient [189], which might have clinical manifestations on MT given that the integrity of the large intestine is considered less susceptible to ischemic injury [82]. Contrary to previous beliefs, the influence of splanchnic reperfusion following exertional-heat stress appears to be an unlikely mechanism of GI barrier integrity loss [82]. Indeed, one study found plasma I-FABP concentrations correlated with splanchnic (stomach) hypoperfusion during moderate intensity exercise (*r* = 0.59), though following post-exercise intestinal reperfusion, I-FABP concentrations began to recover within the first 10 min [86].

Inflammatory cytokines comprise a large family of intercellular pleiotropic signaling molecules that perform many regulatory functions, and are primarily involved in innate immunity [191]. Strenuous exercise induces strong pro-inflammatory (TNF-α, IL-1β, IL-6, IFN-γ), followed by anti-inflammatory (IL-1ra, IL-4, IL-10) responses throughout numerous cells and tissues across the body [192]. The specific biological roles of individual cytokines are incompletely understood and are likely context dependent. That said, several pro-inflammatory cytokines released post-exercise (e.g., TNF-α) appear to disrupt GI barrier integrity [176]. Potential regulatory mechanisms might include: direct modulation of several cell signaling pathways that regulate TJ protein complex stability [193,194,195]; and the indirect pyrogenic modulation of body temperature where local hyperthermia damages the GI barrier [196,197]. With EHS cases, pro-inflammatory cytokines are produced upon immune activation (e.g., nuclear factor kappa-β transcription) following binding between MT products and toll-like receptors located on cell surface membranes [178]. This response appears to operate through a positive feedback loop that may further promote GI MT, cytokine production, and potentially culminate in fatal septic shock [198].

## 6. Nutritional Countermeasures

Nutritional countermeasures could modulate key cellular pathways involved in mitigating exertional-heat stress induced GI barrier integrity loss. Diet regimens and nutrition supplements with evidence they can influence GI barrier integrity following exercise and/or exertional-heat stress will be reviewed. The mechanistic basis of each nutritional intervention, evidence of improved GI barrier function following exercise and practical recommendations are presented (Table 5).

## 7. Carbohydrate

Carbohydrates (CHO) are the main macronutrient of western diets and are an essential energy substrate in sustained moderate and high intensity exercise. The physiological response to CHO ingestion is highly dependent upon its biochemical formula, where high glycemic index CHO (e.g., glucose, maltose) have rapid bioavailability, and low glycemic index CHO (e.g., fructose, galactose) have delayed bioavailability. The volume, tonicity and osmolality of CHO is equally influential. In healthy resting humans, ingestion of a single CHO-rich meal (55–70% of total kilo-calories) evokes equivocal (endotoxin [199,200,201] or slightly improved (I-FABP; [148,149]) GI barrier integrity postprandially. However, rodent experimental models of acute GI distress indicate that oral ingestion of maltodextrin [202] or sucrose [203] favorably influence GI barrier integrity. Mechanisms of action at the whole-body level are likely multifactorial, including regulation of the GI microbiota [204] and an elevation of splanchnic perfusion [205]. Nevertheless, in vivo and in vitro studies indicate that high glucose exposure might reduce GI TJ stability through an abnormal redistribution of several TJ proteins [206]. Compared with ingestion of a single CHO-rich meal, ingestion of a single fat-rich meal results in acute GI MT [200,201,207].

The ingestion of CHO pre-, during and post-exercise in athletic populations, is widely recommended to improve exercise performance [208], accelerate recovery [209] and maintain immune function [210]. In comparison, the influence of CHO on GI barrier integrity has received less attention, despite being associated with the onset of GI complaints [211] and increased splanchnic perfusion [212]. Contrary to proposed hypotheses, preliminary research found no influence of CHO beverage ingestion (30–60 g·h^−1^ glucose), compared with water, on GI barrier integrity (utilising the DSAT) during 60–90 min of moderate intensity exercise (70% VO_2max_) [111,121]. However, follow-up studies reported attenuated GI barrier integrity loss (I-FABP and DSAT) with glucose ingestion (60 g·hour^−1^) during two-hours moderate intensity running (60% VO_2max_) in the heat (35 °C and 25% relative humidity (RH); [129]), and with sucrose ingestion (40 g·h^−1^) prior/during a one-hour moderate intensity cycle (70% watt_max_) [140]. However, neither intervention ameliorated the severity of GI MT. Formulations of single- and multi-transportable CHO mixtures (i.e., 1.8 g·min^−1^ glucose; 1.2 and 0.6 g·min^−1^ glucose plus fructose; 0.6 and 1.2 g·min^−1^ glucose plus sucrose) all tended to (interaction effect *p* = 0.10) reduce I-FABP concentrations (area under the curve at 30 min intervals) to a similar extent relative to water during three hours of low-intensity cycling (50% Watt_max_) [139]. Similarly, ingestion of 60 g·h^−1^ of either potato flesh puree or carbohydrate gel (2:1 maltodextrin/fructose) were able to completely attenuate the rise in I-FABP observed throughout a 2.5 h mixed-intensity cycle (2 h 60% VO_2max_ then a 20 km time trial in temperate conditions) [149]. To date, only one study has reported an adverse effect of CHO ingestion during exercise (1 h 70% VO_2max_ running in 35 °C and 12–20% RH) on GI barrier integrity, with ingestion of a multi-transportable CHO gel (18 g maltodextrin and 9 g fructose) 20-min into exercise shown to increase GI barrier integrity (I-FABP and endotoxin) loss relative to a placebo [213]. Surprisingly, in the placebo condition exertional-heat stress had no influence on GI barrier integrity, whilst in the CHO condition the magnitude of GI integrity loss was minimal. Currently, little is known about the influence of pre-exercise CHO availability on GI barrier integrity. One study reported that 48-h low (20% CHO, 65% fat) versus high (60% CHO, 25% fat) CHO-diet had no influence on GI MT after a laboratory duathlon [175]; whilst a similar study reported no influence of a 24 h low or high FODMAP diet on GI barrier integrity (I-FABP, LBP, sCD14-ST) following 2 h of exertional-heat stress [166].

Practical recommendations for CHO ingestion on GI barrier integrity are unable to be established at present, given the large variation in findings from seemingly comparable studies. This lack of consistency cannot be attributed to differences in prandial state, exercise intensity, CHO type/dose or participant demographic. In general, the application of traditional sports nutrition guidelines for CHO ingestion does not appear to adversely influence GI barrier integrity, and more likely would appear to offer favorable benefits. Future work is required to determine the most effective CHO formulations for fueling exercise and maintaining GI barrier integrity. Factors that may be important include: the carbohydrate source (e.g., potato, maize), dextrose equivalence, osmolarity, sugar profile and delivery format (e.g., drink, gel, energy chew, or bar). The impact of pre-exercise CHO status (e.g., low carbohydrate training, or fasted training) may also influence the GI barrier response to feeding. The strategy of gut-training (i.e., multiple exercise sessions with high [90 g·h^−1^] CHO intake) to improve CHO tolerance during exercise does not appear to strengthen the GI barrier [211].

## 8. Glutamine

Glutamine is the most abundant amino acid in human tissue and plasma, where it performs numerous important regulatory functions. It is a conditionally essential nutrient during states of catabolic stress (e.g., starvation, trauma and severe infection), and is the major energy substrate of GI enterocytes. The use of l-Glutamine supplementation to support GI barrier function has received extensive examination [214]. Benefits have repeatedly been shown in humans following large intravenous _L_-glutamine infusions (~0.2–0.5 g·kg·day^−1^) in patients with critical illness indicative of glutamine deficiency, including severe burns [215,216], post-infectious irritable bowel syndrome [217], and major abdominal trauma [218]. In comparison, benefits are less prominent with low dose oral ingestion (<0.2 g·kg·day^−1^) in chronic GI diseases patients, whom are unlikely to be glutamine deficient and/or exposed to acute stress [219,220]. Mechanisms of action appear multifactorial including: increased epithelial cell proliferation [221]; upregulation of cytoprotective intracellular heat shock protein (I-HSP) expression [222]; modulation of inflammatory signaling pathways [223]; increased vasodilating factors (e.g., nitric oxide); GI microbiota regulation [224]; enhanced GI glutathione status [225]; and improvement in TJ stability through increased expression of multiple TJ proteins [226,227].

Supplementation with l-glutamine is not presently endorsed by sports nutrition guidelines, on the basis of weak evidence demonstrating improved immune function [210] or exercise-performance [228]. Early research investigating the effect of l-Glutamine supplementation on exercise-induced GI permeability (assessed with DSAT), found no additional benefit of co-administering _L_-glutamine (0.018 g·kg^−1^ BM) with CHO (0.18 g·kg^−1^BM) every 10 min during a one-hour moderate-intensity run (70% VO_2max_), in comparison to CHO alone [121]. Unfortunately, l-Glutamine was not assessed in isolation and the total dose consumed was only circa 8–12 g. Since then, researchers have changed their focus from low dose l-glutamine supplementation to maintain circulating concentrations, to provision of large oral doses to saturate the GI tissue prior to exercise. Both chronic (3 × 0.3 g·kg·FFM^−1^ for seven days; [131]) and acute (0.9 g·kg·FFM^−1^ two-hours pre-exercise [116]) l-glutamine ingestion raised circulating concentrations by ~2.5-fold (suggestive of GI saturation) and attenuated the rise in the GI permeability (DSAT ratio) from basal conditions following a one-hour moderate-intensity run (70% VO_2max_) in the heat (30 °C/12–20% RH). Using an identical experimental-design, it was subsequently shown that l-glutamine doses of 0.25, 0.5 and 0.9 g·kg·FFM^−1^ suppressed the post exertional-heat stress rise in serum I-FABP concentration (~0–20%) and DSAT ratio (~25–40%). Although the authors reported a dose-dependent effect on GI barrier integrity [133], statistical significance testing was not undertaken, with these conclusions drawn from magnitude based inference analysis. Recently, ingestion of 0.9 g·kg·FFM^−1^ of _L_-glutamine one hour prior to a 20 km cycling time trial in the heat (35 °C, 50% RH) blunted the rise in circulating post-exercise I-FABP, although this study’s conclusions were drawn from a linear mixed methods Bayesian statistical approach [154].

Practical recommendations support the use of a single L-Glutamine dose (0.90 g·kg·FFM−1) two-hours pre-exercise to protect GI barrier integrity. Given the requirement to only ingest a single acute-dose in the hours prior to exertional-heat stress, the supplementation protocol has clear real-world application in terms of both implementation logistics and expense. Further work is required to confirm these findings following more severe exertional-heat stress protocols and extending analysis to include secondary markers of GI MT. The oral tolerance and safety of such large L-glutamine doses requires clinical assessment as it is above general guidelines (5–10 g) for sports supplements [229]. Likewise, a limitation of all previous research has been the performance of trials in the fasted state, whereby positive findings are potentially attributable to improvement in post-prandial splanchnic perfusion, rather than any benefits directly related to L-glutamine. Indeed, ingesting 15 g·20 min^−1^ of whey protein hydrolysate during a 2-h moderate-intensity (60% VO_2max_) run in the heat (35 °C/30% RH) has also been shown to be highly effective in maintaining GI barrier integrity [129]. Future research should focus on determining if specific amino acid mixtures are as effective, or can even outperform L-glutamine alone, for maintaining GI barrier integrity.

## 9. Bovine Colostrum

Bovine colostrum (BC) is the milk produced by cows during the first 24–48 h post-partum, and its composition markedly differs from milk produced later in lactation [230]. In humans, colostrum provides many health benefits to the neonate, including rapid tissue development and immune defense [231]. BC contains a variety of growth factors (e.g., insulin-like growth factor-1; IGF-1) and immunomodulatory components (e.g., immunoglobulins, cytokines) at higher concentrations than human colostrum [232]. The use of a BC nutritional supplement (liquid and powder) to maintain GI barrier function in healthy adults has been shown to reduce GI permeability post NSAID administration [233], and can blunt systemic elevations in endotoxin following critical illness [234]. These findings are supported by in vitro studies on Caco-2 cells, where BC blunted GI cell apoptosis and increased epithelial resistance during heat exposure [113,235]. Mechanisms of action include: increased epithelial cell proliferation [113,236], upregulation of cytoprotective I-HSP expression [114] and improved TJ stability through a reduction in phosphorylated tyrosine concentrations of occludin and claudin-1 [114].

Supplementation with BC has increased in athletic populations in response to recent evidence of enhanced muscle growth rates [237], blunted exercise-associated immunosuppression [238] and improved exercise performance [239]. More recent investigations have assessed the influence of BC on exercise-induced GI damage. In a series of experiments, 14 days of BC (20 g·day^−1^) halved the 3-fold rise in urinary DSAT ratio and circulating I-FABP concentrations following short-duration (20 min) high-intensity running (80% VO_2max_) [113,114,130]. Whilst these results show promise, such benefits appear attenuated by more demanding exercise protocols. Two comparable studies reported no effect of either a moderate (14 days at 20 g·day^−1^; [240] or high (7 days at 1.7 g·kg·day_−1_ (circa ~120–150 g); [182]) BC dosing on I-FABP concentrations following a fatiguing run in the heat (35–40 °C; 50% RH). Likewise, March et al. [105], using their earlier BC supplementation protocol [130], found only minor (~10%) suppression of I-FABP concentration and a non-significant blunting of circulating Bacteroides DNA following a 1-h run (70% VO_2max_) in the heat (30 °C/60% RH).

Practical recommendations support a BC dose of 20 g.day^−1^ for 14 days to protect the GI tract during moderately demanding exercise, though little-to-no benefits appear likely during more intense exercise. Two days of BC supplementation with the same daily dose offered no protective benefits [163]. Chronic low dose (500 mg·day) BC ingestion improved resting GI permeability (DSAT ratio) in athletes during heavy training [241], but chronic high dose (60 g·day) BC ingestion appeared to increase GI permeability [155]. Further work is required to determine the optimal time-course and BC dose to support GI barrier function. As there are large inter-manufacturer variations in BC formulations, future research should include accurate characterisation of the bioactive components in intervention trials, as these components are likely to have a significant bearing on study findings [242]. No studies have successfully measured the influence of BC on secondary GI MT post-exercise. BC appears to be well-tolerated in healthy individuals in doses up to 60 g·day over several weeks, and although IGF-1 is on the World Anti-Doping Agency banned substance list, it is unlikely BC can result a positive doping control [243].

## 10. Nitric Oxide

The free radicle gas, Nitric Oxide (NO), performs multiple signaling roles in the body. Synthesis occurs through two complementary pathways: the NO synthase (NOS) dependent _L_-arginine pathway; and the NOS independent nitrate (NO_3_), nitrite (NO_2_), NO serial reduction pathway [244]. Supplementation with NO precursors, including l-arginine [245], l-citrulline and inorganic NO3 [246], are all capable of upregulating NO bioavailability across the splanchnic organs. Rodent models show this increase in NO blunts GI histopathological damage and subsequent MT following NSAID ingestion [247], small bowel obstruction [248] and experimentally induced ischemic-reperfusion injury [249,250]. The vasodilatory role of NO in maintaining GI microcirculation appears to be one of the main mechanisms [82], with enhanced antioxidant scavenging [251], constrained neutrophil activation [252] and increased GI TJ protein expression [253] as complementary pathways.

No guidelines exist for l-arginine or l-citrulline supplementation in athletic populations [254], and consensus documents do not support its use to improve oxygen uptake kinetics or exercise performance [255]. Only two studies have investigated the influence of nitric oxide precursors on exercise-induced GI barrier integrity loss. A rodent study found addition of 2% l-arginine to the standard diet (over seven days) prevented a rise in GI barrier loss relative to the control following ~1-h forced running to fatigue in the heat (34 °C) [256]. Similarly in humans, Van Wijck et al. [126] found acute l-citrulline supplementation (10 g given 30 min pre-exercise) successfully maintained splanchnic perfusion and blunted the rise in systemic I-FABP during one hour of moderate intensity cycling (70% watt_max_). However, this intervention did not reduce peak post-exercise I-FABP concentrations, or the urinary DSAT ratio.

Inorganic NO3 supplementation has increased in athletic populations over the last decade [257]. Its popularity is founded upon evidence showing NO_3_ supplementation (~8 m mol, acutely and chronically) reduces the oxygen cost of exercise, enhances muscle efficiency and improves prolonged aerobic performance (10–40 min) [257]. There is limited evidence addressing NO_3_ supplementation and exercise-induced GI barrier integrity loss. One placebo-controlled study found acute sodium NO3 (800 mg given 2.5 h pre-exercise), did not attenuate the rise in circulating I-FABP or LBP concentration following 1-h of moderate intensity cycling (70% watt_max_) [140].

Practical recommendations regarding the use of l-arginine, l-citrulline or inorganic NO3 to protect the GI tract during exercise are inconclusive. Further work is required to substantiate present findings and to verify any benefits over a range of exercise protocols. Likewise, evidence is required to confirm whether benefits are observed in highly trained populations (who tend not to respond to NO supplementation), and to determine which NO precursors provide the most effective GI protection. A further practical consideration is the apparent impaired thermoregulation associated with reduced cutaneous vasodilation, which might disrupt the GI barrier especially when exercising in the heat [258,259].

## 11. Probiotics

Probiotics are live microorganisms considered to regulate the GI microbiota, which might confer health benefits when consumed in adequate quantities [260]. They are found in low concentrations across various food sources (e.g., non-pasteurised dairy products), and regular consumption has been recommended in patients with GI conditions since the early 1900s [260]. More recently, probiotic supplementation to support GI barrier function has received extensive examination. Whilst positive barrier effects are reported in ~50% of human studies, these are not universal, and may reflect the large variations in dose and strains administered [261,262]. Inconclusive effects are also reported in vitro on GI cellular apoptosis and epithelial integrity when Caco-2 cells are cultured with probiotics prior to insult [263,264]. Mechanisms of action are incompletely understood, but are believed to include: inhibition of pathogenic bacterial overgrowth; competition with pathogenic bacteria for binding sites on mucins and/or epithelial cells; increased mucosal immunoglobulin and antimicrobial proteins secretion; increased epithelial cell proliferation; upregulated I-HSP concentrations; suppressed local GI inflammation; and increased TJ stability through upregulation of GI TJ protein expression (for review see: [265]).

Probiotic supplementation is increasingly popular in athletic populations, despite inconsistent effects of their use for either maintaining immune health or improving exercise performance [266]. With respect to GI barrier integrity, four weeks daily consumption of a multi-strain probiotic (45 × 10^9^ colony forming units [CFU]; from three strains) blunted DSAT ratios (8%) and circulating endotoxin concentrations (~12%) following a ~35-min fatiguing run (80% ventilatory threshold) in the heat (35 °C/40% RH) [267]. A follow-up study reported daily ingestion of a similar multi-strain probiotic (3 × 109 CFU; from nine strains) for a period of twelve weeks approximately halved basal endotoxin concentrations immediately prior to and 6-days following an ultra-triathlon [268]. In contrast, seven days high-dose single strain probiotic supplementation (45 × 10^11^ CFU·day Lactobacillus Casei) was associated with an increased rise in endotoxin concentrations, compared with placebo, following two hours moderate-intensity running (60% VO_2max_) in the heat (34 °C/32% RH) [167]. Similarly, the daily ingestion of another single strain probiotic (35 × 10^9^ CFU Bifidobacterium longum) had no effect on resting endotoxin concentrations following six weeks of pre-season training in collegiate swimmers [269]. Likewise, four weeks daily supplementation with a multi-strain probiotic (25 × 10^9^ CFU; from five strains) had no influence on either DSAT, I-FABP or sCD14 responses following a simulated 42.2 km marathon in temperate conditions [134]. Finally, four weeks supplementation with a single strain probiotic (2 × 10^8^ CFU Lactobacillus Salivarius) had no influence on DSAT responses, (or fecal microbial composition), following two hours of moderate intensity running (60% VO_2max_) in temperate conditions [270]. It is unlikely the final two studies were sufficiently powered to detect any influence of probiotic supplementation of GI barrier integrity.

The present data indicate that probiotic supplementation has little for supporting GI barrier integrity in response to exercise. It is not possible to elucidate whether inconsistent responses are attributable to the specific probiotic strain, duration of supplementation or another factor. Future research is required to develop probiotic supplementation regimes and will need to address factors such as strain(s), timing and dose. It will also be necessary to verify potential efficacy using relevant exercise (heat stress) protocols. Global metabolomics approaches have linked exercise-induced GI barrier function loss with alterations in GI microbiota composition during a four-day military arctic training exercise (51 km ski march; [271]), and such methodologies should be applied when developing probiotic supplements to support GI barrier integrity. Probiotic use is considered safe in healthy populations, when consumed acutely and chronically [266].

## 12. Polyphenols

Polyphenols are natural compounds that defend plants against damage from radiation and pathogens. Over 8000 polyphenols have been identified, which are classified into four major groups: flavonoids; phenolic acids; stilbenes; and lignans. Quercetin is the most abundant dietary flavonoid polyphenol [272], and in rodents’ supplementation has been shown to maintain GI barrier integrity [273]. However, in vitro evidence from human Caco-2 cells is less conclusive, with quercetin shown to both improve [267,274] and impair [275,276] GI barrier integrity in response to heat stress. Proposed mechanisms in favorable studies include modulation of vasodilatory factors (e.g., NO [277]), elevated antioxidant scavenging [278] and improved TJ stability through upregulation of several TJ proteins [279]. Proposed mechanisms in non-favorable studies relate to reduced cytoprotective I-HSP expression [280] and TJ stability through disruption in occludin TJ protein localisation [275]. Both positive and negative responses have been comparatively reported when Caco-2 cells are supplemented in vitro with additional polyphenols [277,279]. Human studies assessing polyphenol supplementation efficacy on GI barrier integrity are lacking [277], and where in vitro studies administer physiologically relevant polyphenol doses the effects have been negligible [281].

Polyphenol supplementation is increasingly popular in athletic populations [282]. This is founded upon moderate evidence of enhanced skeletal muscle recovery from micro-damage [283], blunted exercise-associated immunosuppression [284] and in some cases improved (1–3%) endurance exercise performance [285]. With respect to polyphenol supplementation and exercise-induced GI barrier integrity, the effect of daily quercetin supplementation (2 g·day one hour pre-exercise) on GI permeability following the first and seventh days of a standardised isothermic walking (100 min; 1.8 m·s^−1^ in 46 °C/20% RH) heat acclimation regime was assessed [162]. On both days, quercetin ingestion stimulated a ~two-fold rise in urinary lactulose and plasma endotoxin compared with a placebo condition. More promisingly, supplementation with curcumin (3 days of 0.5 g·day), a constituent of turmeric, blunted circulating I-FABP concentrations by ~30% after one-hour moderate intensity running (65% VO_2max_) in the heat (37 °C/25% RH; [156]).

There are no practical recommendations supporting polyphenol use to protect the GI tract during strenuous exercise. Despite promising in vitro observations, more work is required to determine the optimal formulation, time-course and polyphenol dose to support GI barrier function across different exercise-modalities. No studies have successfully measured the effect of polyphenols on secondary GI MT post-exercise and clearly future studies should attempt to control for dietary polyphenol intake.

## 13. Zinc-Carnosine

Zinc-Carnosine (ZnC) is a pharmaceutical chelate of zinc and _L_-carnosine [286]. It is widely used in Japan to treat gastric ulcers [287], and more recently has been marketed in Europe to support GI health [288]. Zinc is an essential trace element and a co-factor in numerous tissue regenerative and immunomodulatory enzymatic reactions [289], whilst _L_-carnosine is a cytoplasmic dipeptide of beta-alanine and _L_-histidine [290]. Daily ZnC ingestion improves GI barrier integrity in healthy humans following chronic GI barrier damaging NSAID ingestion [288,291]. These protective benefits are reported to be synergistic compared with consuming either ingredient individually [292]. In vitro studies of rat intestinal and human Caco-2 cells support these reports, where ZnC blunts GI cellular apoptosis [293,294] and increases epithelial electrical resistance [114] upon damage, in a dose-dependent fashion. In swine, whose GI physiology closely resemble that of humans, short-term (7–17 days) supplementation (60 mg·kg·day^−1^) with a patented zinc amino-acid complex animal feed, reduced GI MT (endotoxin, TNF-α) during 12–14 h of cyclic heat stress [295,296]. Similar benefits are reported on GI barrier integrity (e.g., TJ expression, villus height) following chronic zinc supplementation (~60–75 mg·kg·day^−1^) on other livestock (e.g., broiler chickens, bovine) when raised under cyclic heat stress [297,298]. Mechanisms of action appear multifactorial, including increased: epithelial cell proliferation [291]; I-HSP concentrations [114]; antioxidant activity [299]; and stability of TJs through blunting phosphorylated occludin and claudin-1 expression [114].

No guidelines exist concerning ZnC supplementation in athletic populations. Athletes are recommended to ensure sufficient dietary zinc ingestion (EU RDA = 10 mg·day^−1^) to prevent deficiencies, and to supplement with large oral doses (~75 mg·day^−1^), when suffering from acute upper respiratory tract infection to accelerate recovery [210]. Though _L_-Carnosine supplementation is uncommon, supplementing β-alanine (~65 mg·kg·day^−1^) the rate-limiting precursor for muscle _L_-carnosine synthesis, has been shown to increase muscle carnosine stores [299]. To date, only one study has investigated the influence of ZnC on exercise-induced GI damage. Fourteen days of ZnC (75 mg·day^−1^) attenuated a 3-fold rise is DSAT ratio by 70% after short-duration (20 min) high-intensity running (80% VO_2max_) [114]. This effect was comparable to that observed with BC (20 g·day^−1^ for 14 days) in the same study, and when the two-treatments were combined the benefits appeared synergistic (85% reduction DSAT ratio). Furthermore, the combination of ZnC and BC blunted the exercise-induced increase in DSAT ratio by 30% after only two-days, whilst no protection was offered by either ingredient alone at this point [114].

Practical recommendations support ZnC use at a dose of 75 mg·day^−1^ for 14 days to protect the GI tract during moderately demanding exercise. Further work is needed to substantiate existing findings and verify the potential benefits of ZnC during more strenuous exercise. No studies have successfully measured the influence of ZnC on secondary GI MT post-exercise. Research is required to determine the optimal time-course and dose of ZnC to support GI barrier function with chronic and acute supplementation. Larger doses of ZnC (150 mg·day^−1^) appear well-tolerated in GI disease patients in the short-term [300], and dose-dependent in vitro evidence suggests this might offer greater protection [292]. Co-ingestion of copper with zinc (1:10 ratio or 2 mg·day^−1^) appears to prevent zinc inhibiting copper absorption [210].

## 14. Limitations and Future Directions

Investigation of nutritional countermeasures that support GI barrier integrity during strenuous exercise is an important and expanding area of research. Preliminary observations indicate that some diet regimens and dietary supplements could benefit exercising populations. Optimal supplementation strategies should be safe, well-tolerated, practical (e.g., affordable/low mass), fast acting and effective in a wide range of scenarios (e.g., exercise intensity/duration, population). It is also important that they are without secondary adverse responses, especially those relating to skeletal muscle adaptation, thermoregulation, immune function, bone health etc. Whilst there are numerous examples of well-conducted studies reporting beneficial effects from diet regimens and individual supplements on GI barrier integrity, it is currently not possible to provide definitive guidance. In part this is due to limitations and variations in study designs and in some instance’s incomplete characterisation of the bioactive nutrients.

Future research should address diet regimens/nutritional supplements that satisfy the above requirements when tested in the most demanding scenarios (e.g., high intensity/prolonged exertional-heat stress). It would appear very worthwhile to assess the synergy between ingredients that maintain GI integrity, especially if they are considered to act via different biochemical pathways. Further supplements that warrant future exploration include: omega-3 polyunsaturated fatty acids [301]; vitamin C [171]; vitamin E [132]; vitamin D [157] and prebiotics [302]. Research should target specific populations (e.g., gender, training status, heat-acclimated, GI disease), exercise modalities (especially prolonged duration), supplementation timings (e.g., repeat dosing, delayed/post-exercise ingestion) and monitor the continued efficacy of supplementation following chronic application. Of note, future research is warranted to determine the most damaging exercise protocol on GI barrier, which possibly involves a combination of prolonged/intense exercise performed in the heat.

From a methodological perspective, it is recommended that future studies assess a battery of relevant GI barrier integrity markers (e.g., DSAT, plus I-FABP/I-BABP/claudin-3, plus endotoxin/LBP/sCD14/bactDNA) and monitor alterations in the proposed mechanistic pathways (e.g., splanchnic perfusion, I-HSPs) underpinning any functional benefits. Key extraneous variables should be controlled, including: prandial state [203]; hydration status [144]; beverage temperature [146]; prior NSAID ingestion [120]; habitual diet and supplement use.

## 15. Conclusions

EHS is a life-threatening disease involving thermoregulatory failure, which sporadically arises in otherwise healthy individuals following performance of strenuous exercise or occupationally arduous tasks. Current EHS management policy primarily takes a thermoregulatory management approach despite evidence of MT following loss of GI barrier integrity being an important process in the disease pathophysiology. A range of techniques are available to assess GI barrier integrity in vivo, and a battery approach monitoring multiple measures in both field and research settings is recommended. The severity of GI barrier integrity loss following exertional-heat stress appears to be intensity and duration-dependent, with thermoregulatory strain being an additional risk factor. Considerations for the specific GI barrier integrity assessment technique must be made when interpreting individual study’s conclusions, whereby I-FABP responses typically provided the greatest sensitivity. The specific aetiology of exertional-heat stress induced GI barrier integrity loss is poorly defined, but likely relates to the direct effects of localised hyperthermia, ischemia-reperfusion injury and neuroendocrine-immune alterations.

A range of nutritional countermeasures have been shown to positively affect GI barrier integrity following strenuous exercise and exercise-heat stress. However, despite rapid advancements in this field, definitive recommendations cannot be provided due to the heterogeneity of experimental designs. Nevertheless, promising effects have been associated with following general sports nutrition CHO supplementation guidelines during exercise (30–100 g·h^−1^ liquid multi-transportable CHO), and acute L-glutamine ingestion two hours pre-exercise (0.25–0.9 g·kg·FFM^−1^). Benefits from BC, and probiotics likely relate to the specific supplement formulation, and hence require further investigation. Despite a sound rationale for the use of NO precursors and polyphenols to limit exercise-induced GI barrier integrity loss, substantive supporting evidence is currently absent. ZnC requires further verification, where short-term (1–3 days) high-dose supplementation appears an attractive consideration. Further well-controlled research in nascent areas could elucidate potential treatment options for exercise-induced GI barrier integrity loss.

## Figures and Tables

**Figure 1 nutrients-12-00537-f001:**
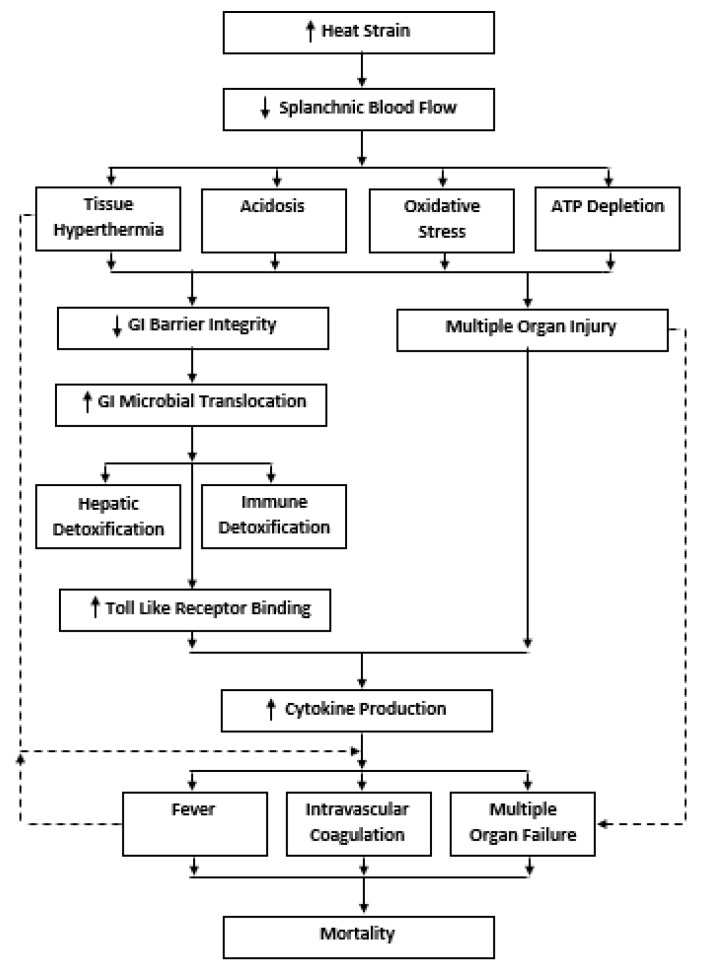
The gastrointestinal paradigm of exertional heat stroke.

**Table 1 nutrients-12-00537-t001:** Overview of in vivo techniques to assess GI barrier integrity.

Technique	Sample	Method	Site	Limitations
**Active Techniques**				
Dual-Sugar Absorption Test (DSAT)	Urine or blood	HPLC (+) MS	Small GI Integrity	Gold-standard. High reliability. Time-consuming (5 h urine, >2.5 h blood). No standard protocol with exercise. Well studied.
Multi-Sugar Absorption Test (MSAT)	Urine or blood	HLPC (+) MS	Entire GI Integrity	Gold-Standard. Segmental GI integrity. Time-consuming (5 h urine, >2.5 h blood). No standard protocol with exercise. Few studies.
Polyethylene Glycol (PEG) Absorption Test	Urine	HLPC (+) MS	Entire GI Integrity	Validated against MSAT. Can include multiple weight PEGs (e.g., 100, 400, 1000, 4000 kDa). Time-consuming (5 h urine). Few studies.
**Passive Techniques**				
Intestinal Fatty Acid Binding Protein (I-FABP)	Urine or Blood	ELISA	Epithelial injury	Tissue specific (duodenum and jejunum). Short half-life (11 min). Weak correlations with DSAT. Well studied.
Ileal Bile-Acid Binding Protein (I-BABP)	Urine or Blood	ELISA	Epithelial injury	Tissue specific (ileum). Few studies. Weak correlations with I-FABP. Few studies.
Diamine Oxidase (DAO), α-Glutathione s-Transferase (α-GST), Smooth Muscle 22 (SM22)	Blood	ELISA	Epithelial injury	Non-tissue specific. Few studies.
Claudin-3 (CLDN3)	Urine or Blood	ELISA	TJ Integrity	Non-tissue specific. Few studies.
Zonulin	Blood or Faeces	ELISA	TJ Integrity	Non-tissue specific. Assay cross-reactivity (complement C3). Moderate studies.
Endotoxin (LPS)	Blood	LAL assay	MT	Tissue specific. Sample contamination causes false-positives. Hepatic removal and receptor binding cause false-negatives. Well studied.
LPS Binding Protein (LBP)	Blood	ELISA	MT	Tissue specific. Lower risk of false-positives than endotoxin. Indirect marker of endotoxin exposure. Influenced by hepatic production. Long half-life (12–14 h). Few studies.
Soluable-CD14 (sCD14-ST)	Blood	ELISA	MT	Tissue specific. Lower risk of false positives than endotoxin. Influenced by hepatic production and monocytes shedding. Few studies.
D-lactate	Blood	ELISA	MT	Predominately tissue specific. Potentially influenced by methylglyoxal metabolism. Few studies.
16s Bacterial rDNA (bactDNA)	Blood	Real-time PCR assay	MT	Tissue specific. Novel. Lower risk of false-positives than endotoxin. Few studies.

Abbreviations: HPLC, high performance liquid chromatography; MS, mass spectrometry; ELISA, enzyme-linked immunosorbent assay; LAL, limulus amoebocyte lysate assay; PCR, polymerase chain reaction.

**Table 2 nutrients-12-00537-t002:** Influence of acute exercise-(heat) stress on small-intestine DSAT responses.

Author	Subjects	Exercise Protocol	Peak T_Core_ (°C)	Mean HR (bpm)	Biofluid, DSAT L/R or L/M (Timepoint)
van Nieuwenhoven et al. [110]	10 male (MT)	90 min cycling at 70% Watt_max_ (fasted) in T_amb_ 19 °C (RH = N/A)	N/A	N/A	Urine L/R (5 h): 0.007 ^s^
van Nieuwenhoven et al. [117]	10 male (MT)	90 min cycling at 70% Watt_max_ (fasted) in T_amb_ 19 °C (RH = N/A)	38.8	N/A	Urine L/R (5 h): 0.008 ^nb,c^
Nieman et al. [107]	20 male and female (UT)	45 min walking uphill (5% grade) at 60% VO_2max_ (fasted) in T_amb_ not reported	N/A	132	Urine L/R (5 h): 0.009 ^nb,c^
Smetanka et al. [122]	8 male (HT)	Chicago marathon (42.2 km) in T_amb_ (fed) 22 °C (48% RH)	N/A	N/A	Urine L/R (5 h): 0.020 ^ns^
Shing et al. [125]	10 male (HT)	~33 min running to fatigue at 80% VE (fed) in T_amb_ 35 °C (40% RH)	39.4	172	Urine L/R (5 h): 0.022 ^nb,c^
Janssen-Duijghuijsen et al. [109]	11 male (HT)	90 min cycling at 50% watt_max_ (fed) in T_amb_ not reported following a sleep-low glycogen depletion regime	N/A	N/A	Urine L/R (5 h): ~0.022 ^ns^ Plasma L/R (1 h): ~0.110 ^s^
Snipe et al. [115,124]	6 male and 4 female (MT)	120 min running at 60% VO_2max_ (fed) in T_amb_ 22 °C (44% RH)	38.5	~150	Urine L/R (5 h): 0.025 ^nb^
Snipe et al. [124]	6 male and 4 female (MT)	120 min running at 60% VO_2max_ (fed) in T_amb_ 30 °C (25% RH)	38.6	~155	Urine L/R (5 h): 0.026 ^nb^
van Wijck et al. [126]	10 male (MT)	60 min cycling at 70% watt_max_ (fasted) in T_amb_ not reported	N/A	N/A	Urine L/R (2 h): 0.027 ^nb,c^
Buchman et al. [127]	17 male and 2 female	Competitive Marathon (fed) in T_amb_ 2 °C with freezing rain	N/A	N/A	Urine L/R (6 h): 0.030 ^ns,c^
Ryan et al. [119]	7 males (MT)	60 min running at 68% VO_2max_ (fasted) in T_amb_ not reported	N/A	N/A	Urine L/M (6 h): 0.029 ^ns^
van Nieuwenh-oven et al. [112]	9 male and 1 female (MT)	90 min cycling at 70% Watt_max_ (fasted) in T_amb_ 19 °C (RH = N/A)	N/A	N/A	Urine L/R (5 h): 0.030 ^ns^
van Wijck et al. [123]	9 male (MT)	60 min cycling at 70% watt_max_ (fasted) in T_amb_ not reported	N/A	N/A	Urine L/R (2 h): 0.030 ^s,c^
Pugh et al. [88]	11 male (MT-HT)	18 × 400 m sprint at 120% VO_2max_ (fed) in T_amb_ not reported	N/A	N/A	Urine L/R (2h): 0.030 ^ns^ Serum L/R (2 h): ~0.051 ^s^
Snipe and Costa [128]	11 male (MT)	120 min running at 60% VO_2max_ (fed) in T_amb_ 35 °C (25% RH)	39.1	~150	Urine L/R (5 h): 0.030 ^nb^
Snipe and Costa [128]	13 female (MT)	120 min running at 60% VO_2max_ (fed) in T_amb_ 35 °C (25% RH)	38.8	~155	Urine L/R (5 h): 0.028 ^nb^
Snipe et al. (Part B) [115]	6 male and 4 female (MT)	120 min running at 60% VO_2max_ (fed) in T_amb_ 35 °C (26% RH)	39.6	~170	Urine L/R (5 h): 0.032 ^nb^
Snipe et al. [129]	6 male and 5 female (MT)	120 min running at 60% VO_2max_ (fed) in T_amb_ 35 °C (30% RH)	39.3	159	Urine L/R (5 h): 0.034 ^nb,c^
March et al. [130]	9 male (MT)	20 min running at 80% VO_2peak_ (fasted) in T_amb_ 22 °C (37% RH)	38.4	170	Urine L/R (5 h): 0.035 ^s,c^
Pals et al. (Part A) [108]	5 male and 1 female (MT)	60 min running at 40% VO_2peak_ (fasted) in T_amb_ 22 °C (50% RH)	38.0	N/A	Urine L/R (5 h): 0.036 ^ns^
Marchbank et al. [113]	12 male (MT)	20 min running to fatigue at 80% VO_2max_ (fasted) in T_amb_ not reported	38.3	N/A	Urine L/R (5 h): 0.038 ^s,c^
van Nieuwenh-oven et al. [111]	9 male and 1 female (MT)	90 min running at 70% VO_2max_ (fasted) in T_amb_ 19 °C (RH = N/A)	N/A	N/A	Urine L/R (5 h): 0.040 ^s^
van Wijck et al. [86]	6 male (HT)	60 min cycling at 70% watt_max_ (fasted) in T_amb_ not reported	N/A	N/A	Urine L/R (5 h): 0.040 ^ns^ Plasma L/R (2.4 h): 0.060 ^s^
Lambert et al. (Part A) [118]	11 male and 9 female (MT)	60 min running at 70% VO_2max_ (fasted) in T_amb_ 22 °C (48% RH)	38.5	N/A	Urine L/R (5 h): 0.049 ^ns,c^
Lambert et al. [121]	13 male and 4 female (HT)	60 min running at 70% VO_2max_ (fasted) in T_amb_ 22 °C (48% RH)	38.3	N/A	Urine L/R (5 h): 0.050 ^nb,c^
Zuhl et al. [131]	4 male and 3 female (LT/MT)	60 min running at 70% VO_2max_ (fasted) in T_amb_ 30 °C (12–20% RH)	39.4	N/A	Urine L/R (5 h): 0.060 ^nb,c^
Zuhl et al. [116]	2 male and 5 female (LT/MT)	60 min running at 70% VO_2max_ (fasted) in T_amb_ 30 °C (12–20% RH)	39.5	N/A	Urine L/R (5 h): 0.060 ^nb,c^
Lambert et al. (Part B) [118]	11 male and 9 female (MT)	60 min running at 70% VO_2max_ (fasted) in T_amb_ 22 °C (48% RH) without fluid ingestion	38.5	N/A	Urine L/R (5 h): 0.063 ^s,c^
Pals et al. (Part B) [108]	5 male and 1 female (MT)	60 min running at 40% VO_2peak_ (fasted) in T_amb_ 22 °C (50% RH)	38.7	N/A	Urine L/R (5 h): 0.064 ^ns^
Lambert et al. [120]	8 male (MT)	60 min running at 70% VO_2max_ (fasted) in T_amb_ 22 °C (48% RH)	38.3	N/A	Urine L/R (5 h): 0.065 ^nb,c^
Buchman et al. [132]	15 male and female (LT-HT)	Road marathon (42.2 km) (fed) in T_amb_ not reported	N/A	N/A	Urine L/M (6 h): 0.070 ^ns,c^
Pugh et al. [133]	10 male (MT)	60 min at 70% VO_2max_ running (fasted) in T_amb_ 30 °C (4–45% RH)	38.5	82.5% of max	Serum L/R (2 h): ~0.080 ^s,c^
Pugh et al. [134]	10 male and 2 female (MT)	42.4 km track marathon (247 ± 47 min; fed) in T_amb_ 16–17 °C (N/A RH)	N/A	~160	Serum L/R (1 h) 0.081 (37%) ^s,c^
Lambert et al. [135]	12 female (LT-HT)	Hawaii Ironman (fed) in T_amb_ not reported	N/A	N/A	Urine L/R (5 h): 0.087 ^nb^
Davison et al. [114]	8 male (MT/HT)	20 min running to fatigue at 80% VO_2max_ (fasted) in T_amb_ not reported	39.3	~170	Urine L/R (5 h): 0.098 ^s,c^
Janssen-Duijghuijsen et al. [136]	4 male and 6 female (LT)	60 min cycling at 70% watt_max_ (fed) in T_amb_ not reported	N/A	N/A	Plasma L/R (1 h): ~0.100 ^s^
Lambert et al. [135]	29 male (LT-HT)	Hawaii Ironman (fed) in T_amb_ not reported	N/A	N/A	Urine L/R (5 h): 0.105 ^nb^
Pals et al. (Part C) [108]	5 male and 1 female (MT)	60 min running at 40% VO_2peak_ (fasted) in T_amb_ 22 °C (50% RH)	39.6	N/A	Urine L/R (5 h): 0.107 ^s^

LT = Low-trained (35–49 mL·kg·min^−1^ VO_2max_); MT = Moderate-trained (50–59 mL·kg·min^−1^ VO_2max_); HT = High-trained (60+ mL·kg·min^−1^ VO_2max_). s = significant change post-exercise (*p* < 0.05); ns = non-significant change post-exercise (*p* > 0.05); nb = no baseline resting data to compare against; c = control/placebo trial of study.

**Table 3 nutrients-12-00537-t003:** Influence of acute exercise-(heat) stress on systemic I-FABP concentrations.

Author	Subjects	Exercise Protocol	Peak T_Core_ (°C)	Mean HR (bpm)	I-FABP (Δ Pre-to-Post Exercise)
Janssen-Duijghuijsen et al. [109]	11 male (HT)	90 min cycling at 50% watt_max_ (fed) in T_amb_ not reported following a “sleep-low” glycogen depletion regime	N/A	N/A	~−90 pg·mL^−1^ (~−65%) ^c^
Kartaram et al. (Part A) [138]	15 male (MT)	60 min cycling at 50% watt_max_ (fed) in T_amb_ not reported	N/A	N/A	~-50 pg·mL^−1^ (~−10%) ^ns^
Lee and Thake (Part A) [137]	7 male (MT)	60 min cycling at 50% VO_2max_ (fed) in T_amb_ 18 °C (35% RH) on day one of temperate acclimation	37.9	133	28 pg·mL^−1^ (8%) ^ns,c^
Trommelen et al. [139]	10 male (HT)	180 min cycling at 50% watt_max_ (fasted) in T_amb_ 18–22 °C (55–65% RH)	N/A	N/A	N/A pg·mL^−1^ (20%) ^ns,c^
Edinburgh et al. (Part A) [148]	12 male (MT)	60 min cycling at 50% VO_2max_ (fed) in T_amb_ 18 °C (35% RH)	N/A	N/A	70 pg·mL^−1^ (34%) ^s^
Edinburgh et al. (Part B) [148]	12 male (MT)	60 min cycling at 50% VO_2max_ (fasted) in T_amb_ 18 °C (35% RH)	N/A	N/A	88 pg·mL^−1^ (20%) ^s^
Osborne et al. (Part A) [142]	8 male (MT-HT)	30 min cycling at 50/70% Watt_max_, then 30 min at 50% watt_max_ (fasted) in T_amb_ 20 °C (55% RH)	38.5	139	138 pg·mL^−1^ (29%) ^ns^
Salvador et al. 2019 [149]	12 male (MT-HT)	120 min cycling at 60% VO_2max_ (fed) then 30–40 min (20 km) time trial in T_amb_ not reported	37.9	~168	N/A pg·mL^−1^ (~50%) ^s, c^
van Wijck et al. [126]	10 male (MT)	60 min cycling at 70% watt_max_ (fasted) in T_amb_ not reported	N/A	N/A	153 pg·mL^−1^ (72%) ^s^
Nava et al. [150]	7 male and 4 female (LT-MT)	56 min mixed intensity (~55% VO_2max_) discontinuous firefighting exercises (fed) in T_amb_ 38 °C (35% RH) on day one of two	38.7	~161	~160 pg·mL^−1^ (23%) ^ns, c^
Van Wijck et al. [123]	9 male (MT)	60 min cycling at 70% watt_max_ (fasted) in T_amb_ not reported	N/A	N/A	179 pg·mL^−1^ (61%) ^s^
Lee et al. (Part C) [137]	7 male (MT)	60 min cycling at 50% VO_2max_ (fed) in T_amb_ 18 °C (35% RH) and F_i_O_2_ = 0.14% on day one of hypoxic acclimation	38.2	149	193 pg·mL^−1^ (43%) ^s,c^
Lis et al. [151]	13 male and female (MT)	45 min cycling at 70% watt_max_ and 15 min cycling time trial (fed) in 20 °C (40% RH)	N/A	168	210 pg·mL^−1^ (223%) ^s,c^
Pugh et al. [134]	10 male (MT)	60 min at 70% VO_2max_ running (fasted) in T_amb_ 30 °C (4–45% RH)	38.5	82.5% of HR max	250 pg·mL^−1^ (71%) ^s,c^
Snipe et al. (Part A) [115,124]	6 male and 4 female (MT)	120 min running at 60% VO_2max_ (fed) in T_amb_ 22 °C (44% RH)	38.5	~150	274 pg·mL^−1^ (127%) ^s^
Sheahen et al. (Part A) [145]	12 male (MT)	45 min running at 70% VO_2max_ (fasted) in T_amb_ 20 °C (40% RH)	38.2	165	281 pg·mL^−1^ (49%) ^s^
Lee et al. (Part B) [137]	7 male (MT)	60 min cycling at 50% VO_2max_ (fed) in T_amb_ 40 °C (25% RH) on day one of heat acclimation	38.7	151	282 pg·mL^−1^ (76%) ^s,c^
Morrison et al. (Part B) [152]	8 male (UT)	30 min cycling at 50% heart rate reserve (HRR), 30 min jogging at 80% HRR and 30 min running time trial (fed) in T_amb_ 30 °C (50% RH)	38.6	N/A	283 pg·mL^−1^ (276%) ^s,c^
Barberio et al. [72]	9 male (MT)	~24 min running at 78% VO_2max_ (fed) in T_amb_ 40 °C (40% RH) prior to heat acclimation	39.0	N/A	297 pg·mL^−1^ (46%) ^s,c^
Hill et al. [143]	10 male (MT)	60 min running at 65% VO_2max_ (fasted) in T_amb_ not reported	N/A	~170	300 pg·mL^−1^ (50%) ^ns,c^
van Wijck et al. [86]	15 male (HT)	60 min cycling at 70% watt_max_ (fasted) in T_amb_ not reported	N/A	N/A	306 pg·mL^−1^ (61%) ^s^
Kashima et al. [153]	5 male and 3 female (MT)	30 intermittent 20 s cycle sprints at 120% watt_max_, with 40 s recovery between each (fed) in 23 °C (40% RH)	N/A	150	343 pg·mL^−1^ (266%) ^s^
Pugh et al. [88]	11 male (MT-HT)	18 × 400 m sprint at 120% VO_2max_ (fed) in T_amb_ not reported	N/A	N/A	348 pg·mL^−1^ (72%) ^s^
March et al. [130]	9 male (MT)	20 min running at 80% VO_2peak_ (fasted) in T_amb_ 22 °C (37% RH)	38.4	170	350 pg·mL^−1^ (61%) ^s,c^
Janssen-Duijghuijsen et al. [136]	4 male and 6 female (LT)	60 min cycling at 70% watt_max_ (fed) in T_amb_ not reported	N/A	N/A	~350 pg·mL^−1^ (~77%) ^s,c^
Sheahen et al. (Part B) [145]	12 male (MT)	45 min running at 70% VO_2max_ (fasted) in T_amb_ 30 °C (40% RH)	38.3	163	369 pg·mL^−1^ (63%) ^s^
Costa et al. [144]	11 male (MT-HT)	120 min running at 70% VO_2max_ (fed) in T_amb_ 25 °C (35% RH)	N/A	148	371 pg·mL^−1^ (86%) ^ns,c^
Osborne et al. [154]	12 male (MT-HT)	33 min (20 km) cycling time trial (fasted) in 35 °C (50% RH)	39	167	441 pg·mL^−1^ (83%) ^s,c^
Kartaram et al. (Part B) [138]	15 male (MT)	60 min cycling at 70% watt_max_ (fed) in T_amb_ not reported	N/A	N/A	~500 pg·mL^−1^ (~66%) ^s^
Kartaram et al. (Part C) [138]	15 male (MT)	60 min cycling at 85/55% watt_max_ (fed) in T_amb_ not reported	N/A	N/A	~500 pg·mL^−1^ (~66%) ^s^
McKenna et al. [155]	10 male (MT)	46 min running at 95% VE threshold (fasted) in T_amb_ 40 °C (50% RH)	39.7	N/A	516 pg·mL^−1^ (52%) ^s,c^
Karhu et al. [141]	17 male (MT-HT)	90 min running at 80% of best 10 km race time (fed) in T_amb_ not reported	N/A	N/A	531 pg·mL^−1^ (151%) ^s^
Snipe and Costa [146]	6 male and 6 female (MT)	120 min running at 60% VO_2max_ (fed) in T_amb_ 30 °C (35% RH)	38.8	160	573 pg·mL^−1^ (184%) ^s,c^
Snipe et al. (Part B) [124]	6 male and 4 female (MT)	120 min running at 60% VO_2max_ (fed) in T_amb_ 30 °C (25% RH)	38.6	~155	~580 pg·mL^−1^ (184%)
Hill et al. [143]	10 male (MT)	60 min running at 65% VO_2max_ (fasted) in T_amb_ not reported (F_i_O_2_ = 13.5%)	N/A	~170	700 pg·mL^−1^ (168%) ^ns,c^
Osborne et al. (Part B) [142]	8 Male (MT-HT)	30 min cycling at 50/70% Watt_max_, then 30 min at 50% watt_max_ (fasted) in T_amb_ 35 °C (53% RH)	39.5	159	608 pg·mL^−1^ (140%) ^s^
Szymanski et al. [156]	6 male and 2 female (LT/MT)	60 min running at 68% VO_2max_ (fasted) in T_amb_ 37 °C (25% RH)	39.0	174	800 pg·mL^−1^ (87%) ^s,c^
Morrison et al. (Part A) [152]	7 male (HT)	30 min cycling at 50% heart rate reserve (HRR), 30 min jogging at 80% HRR and 30 min running time trial (fed) in T_amb_ 30 °C (50% RH)	38.6	N/A	806 pg·mL^−1^ (663%) ^s,c^
Snipe et al. [129]	6 male and 5 female (MT)	120 min running at 60% VO_2max_ (fed) in T_amb_ 35 °C (30% RH)	39.3	159	897 pg·mL^−1^ (288%) ^s,c^
Snipe et al. (Part B) [115]	6 male and 4 female (MT)	120 min running at 60% VO_2max_ (fed) in T_amb_ 35 °C (26% RH)	39.6	~170	1230 pg·mL^−1^ (432%) ^s^
Pugh et al. [134]	10 male and 2 female (MT)	42.4 km track marathon (247 ± 47 min; fed) in T_amb_ 16-17 °C (N/A RH)	N/A	~160	1246 pg·mL^−1^ (371%) ^s, c^
March et al. [105]	12 male (MT)	60 min running at 70% VO_2max_ (fasted) in T_amb_ 30 °C (60% RH)	39.3	170	1263 pg·mL^−1^ (407%) ^s, c^
Snipe and Costa [157]	11 male (MT)	120 min running at 60% VO_2max_ (fed) in T_amb_ 35 °C (25% RH)	39.1	~150	1389 pg·mL^−1^ (479%) ^s^
Snipe et al. [158]	13 female (MT)	120 min running at 60% VO_2max_ (fed) in T_amb_ 35 °C (25% RH)	38.8	~155	1445 pg·mL^−1^ (479%) ^s^
Jonvik et al. [140]	16 male (HT)	60 min cycling at 70% watt_max_ (fasted) in T_amb_ not reported	N/A	N/A	1745 pg·mL^−1^ (249%) ^s^
Gaskell et al. [134]	10 male and 8 female (MT-HT)	120 min running at 60% VO_2max_ (fed) in T_amb_ 35 °C (25% RH)	38.6	~151	1805 pg·mL^−1^ (710%) ^s, c^

LT = Low-trained (35–49 mL·kg·min^−1^ VO_2max_); MT = Moderate-trained (50–59 mL·kg·min^−1^ VO_2max_); HT = High-trained (60+ mL·kg·min^−1^ VO_2max_). s = significant change post-exercise (*p* < 0.05); ns = non-significant change post-exercise (*p* > 0.05); c = control/placebo trial of study.

**Table 4 nutrients-12-00537-t004:** Influence of acute exercise-(heat) stress on systemic gastrointestinal microbial translocation responses.

Author	Subjects	Exercise Protocol	Peak T_Core_ (°C)	Mean HR (bpm)	Endotoxin (Δ Pre-to-Post Exercise)
Antunes et al. [169]	19 male (MT)	56 ± 7 min cycling at 90% of first ventilatory threshold (fasted) in 22.1 °C (55% RH)	N/A	^141^	−3 pg·mL^−1^ (−3%) ^ns^
Yeh et al. (Part B) [147]	15 male and 1 female (LT)	60 min running at 70% VO_2max_ (fed) in T_amb_ 22 °C (66% RH)	38.4	~145	−1.1 pg·mL^−1^ (−10%) ^ns^
Zuhl et al. [116]	2 male and 5 female (LT/MT)	60 min running at 70% VO_2max_ (fasted) in T_amb_ 30 °C (12–20% RH)	39.5	N/A	−0.2 pg·mL^−1^ (−7%) ^ns,c^
Osborne et al. (Part A) [142]	8 Male (MT-HT)	30 min cycling at 50/70% Watt_max_, then 30 min at 50% watt_max_ (fasted) in T_amb_ 20 °C (55% RH)	38.5	165	0.1 pg·mL^−1^ (1%) ^ns,#^
Osborne et al. (Part B) [142]	8 Male (MT-HT)	30 min cycling at 50/70% Watt_max_, then 30 min at 50% watt_max_ (fasted) in T_amb_ 35 °C (53% RH)	39.5	182	0.2 pg·mL^−1^ (1%) ^s,#^
Karhu et al. [141]	17 males (MT-HT)	90 min running at 80% of best 10 km race time (fed) in T_amb_ not reported	N/A	N/A	0.3 pg·mL^−1^ (~1%) ^ns,c^
Kuennen et al. [162]	8 male (MT)	100 min walking (6.3 km·h^−1^) at 50% VO_2max_ (fasted) in T_amb_ 46.5 °C (20% RH)	39.3	N/A	~0.5 pg·mL^−1^ (10%) ^ns,c^
Ng et al. [73]	30 males (HT)	Half-marathon (fed) in T_amb_ 27 °C (84% RH)	40.7	172	0.6 pg·mL^−1^ (32%) ^s^
Jeukendrup et al. [163]	29 male and 1 female (HT)	Ironman (3.8 km swim; 185 km cycle; 42.2 km run) (fed) in T_amb_ 9–32 °C	N/A	N/A	1.7 pg·mL^−1^ (666%) ^s^
Guy et al. [170]	20 male (LT-MT)	10 min cycling at 50%, 60%, and 70% watt_max_, then 5 km (fasted) in T_amb_ 35 °C (70% RH)	38.9	160	2 pg·mL^−1^ (9%) ^ns^
Selkirk et al. (Part B) [125]	12 male (HT)	To fatigue (~122 min) uphill walk at 4.5 km.h^−1^ (fasted) in T_amb_ 40 °C (30% RH)	39.7	156	~3 pg·mL^−1^ (200%) ^s^
Shing et al. [165]	10 male (HT)	~33 min running to fatigue at 80% VE (fed) in T_amb_ 35 °C (40% RH)	39.4	172	4 pg·mL^−1^ (15%) ^s^
Snipe et al. (Part A) [115,124]	6 male and 4 female (MT)	120 min running at 60% VO_2max_ (fed) in T_amb_ 22 °C (44% RH)	38.5	~150	4.1 pg·mL^−1^ (5%) ^ns^
Yeh et al. (Part B) [147]	15 male and 1 female (LT)	60 min running at 70% VO_2max_ (fed) in T_amb_ 33 °C (50% RH)	39.3	~145	5 pg·mL^−1^ (54%) ^s^
Antunes et al. (Part B) [169]	19 male (MT)	45 ± 18 min cycling at midpoint between first and second ventilatory threshold (fasted) in 22.1 °C (55% RH)	N/A	^162^	5 pg·mL^−1^ (7%) ^ns^
Antunes et al. (Part C) [169]	19 male (MT)	10 ± 9 min cycling at midpoint between second ventilatory threshold and maximal aerobic power (fasted) in 22.1 °C (55% RH)	N/A	^180^	6 pg·mL^−1^ (5%) ^ns^
Ashton et al. [171]	10 males (LT)	VO_2max_ test (~15 min) on cycle ergometer (fasted) in T_amb_ not reported	N/A	N/A	9.4 pg·mL^−1^ (72%) ^s^
Snipe et al. (Part B) [115]	6 male and 4 female (MT)	120 min running at 60% VO_2max_ (fed) in T_amb_ 35 °C (26% RH)	39.6	~170	9.8 pg·mL^−1^ (11%) ^s^
Gill et al. [167]	8 male (MT-HT)	120 min running at 60% VO_2max_ (fed) in T_amb_ 32 °C (34% RH)	38.6	165	10 pg·mL^−1^ (4%) ^ns, c^
Snipe et al. [129]	6 male and 5 female (MT)	120 min running at 60% VO_2max_ (fed) in T_amb_ 35 °C (30% RH)	39.3	159	10 pg·mL^−1^ (N/A%) ^nb^
Selkirk et al. (Part A) [125]	11 male (LT-MT)	To fatigue (~106 min) uphill walk at 4.5 km.h^−1^ (fasted) in T_amb_ 40 °C (30% RH)	39.1	164	~10 pg·mL^−1^ (300%) ^s^
Lim et al. (Part B) [168]	9 male (HT)	To fatigue (time not given) at 70% VO_2max_ (fed) in T_amb_ 35 °C (40% RH)	39.5	N/A	13 pg·mL^−1^ (92%) ^s,c^
Guy et al. [172]	8 male (LT)	10 min cycling at 50%, 60%, and 70% watt_max_, then 5 km (fasted) in T_amb_ 35 °C (70% RH)	38.6	161	16 pg·mL^−1^ (9%) ^ns,c,#^
Gill et al. [71]	13 male and 6 female (HT)	Multistage ultra-marathon stage 1 (37 km) (fed) in T_amb_ 32–40 °C (32–40% RH)	N/A	N/A	40 pg·mL^−1^ (14%) ^s^
Barberio et al. [72]	9 male (MT)	~24 min running at 78% VO_2max_ (fed) in T_amb_ 40 °C (40% RH) prior to heat acclimation	39.0	N/A	40 pg·mL^−1^ (57%) ^s,c^
Moss et al. [173]	9 male (HT)	45 min cycling at 40% PPO (unstated prandial state) in T_amb_ 40 °C (50% RH) prior to heat acclimation	38.9	153	52 pg·mL^−1^ (27%) ^s,c^
Costa et al. [144]	11 male (MT-HT)	120 min running at 70% VO_2max_ (fed) in T_amb_ 25 °C (35% RH)	N/A	148	96 pg·mL^−1^ (46%) ^ns,c,#^
Gill et al. [164]	14 male and 3 female (HT)	24 h ultramarathon (fed) in T_amb_ 0-20 °C (54–82% RH)	N/A	N/A	122 pg·mL^−1^ (37%) ^s,#^
Machado et al. (Part A) [174]	9 male (MT)	60 min running at 50% VO_2max_ (fasted) in T_amb_ not reported	N/A	N/A	130 pg·mL^−1^ (33%) ^ns,#^
Machado et al. (Part B) [174]	9 male (MT)	60 min running at 50% VO_2max_ (fasted) in T_amb_ not reported (FIO_2_ = 13.5%)	N/A	N/A	250 pg·mL^−1^ (48%) ^s,#^
Gaskell et al. [166]	10 male and 8 female (MT-HT)	120 min running at 60% VO_2max_ (fed) in T_amb_ 35 °C (25% RH)	38.6	~151	LBP ~−2 µg·mL^−1^ (N/A%) ^ns,c^
Selkirk et al. (Part A) [165]	11 male (HT)	To fatigue (~163 min) uphill walk at 4.5 km.h^−1^ (fasted) in T_amb_ 40 °C (30% RH)	39.1	164	LBP ~0 µg·mL^−1^ (0%) ^ns^
Moncada-Jiminez et al. [175]	11 male (MT-HT)	135-min laboratory duathlon at 71% VO_2max_ (15km run and 30km cycle) (fasted) in T_amb_ not reported	38.5	N/A	LBP ~0.59 µg·mL^−1^ (22%) ^s,c^
Selkirk et al. (Part B) [166]	12 male (LT-MT)	To fatigue (~106 min) uphill walk at 4.5 km.h^−1^ (fasted) in T_amb_ 40 °C (30% RH)	39.7	156	LBP ~1.5 µg·mL^−1^ (15%) ^s^
Jonvik et al. [140]	16 male (HT)	60 min cycling at 70% watt_max_ (fasted) in T_amb_ not reported	N/A	N/A	LBP 1.6 µg·mL^−1^ (13%) ^s^
Costa et al. [144]	11 male (MT-HT)	120 min running at 70% VO_2max_ (fed) in T_amb_ 25 °C (35% RH)	N/A	148	sCD14-ST 0.05 µg·mL^−1^ (N/A%) ^ns,c^
Gaskell et al. [166]	10 male and 8 female (MT-HT)	120 min running at 60% VO_2max_ (fed) in T_amb_ 35 °C (25% RH)	38.6	~151	sCD14-ST 0.1 µg·mL^−1^ (N/A%) ^s,c^
Stuempfle et al. [158]	15 male and 5 female (MT)	161-km ultramarathon (26.8 ± 2.4 h; fed) in T_amb_ 0–30 °C (N/A RH)	38.3	N/A	sCD14-ST 0.6 µg·mL^−1^ (63%) ^s^
Pugh et al. [134]	10 male and 2 female (MT)	42.4 km track marathon (4.1 ± 0.8 h; fed) in T_amb_ 16–17 °C (N/A RH)	N/A	~160	sCD14-ST 5.4 µg·mL^−1^ (164%) ^s,c^

LT = Low-trained (35–49 mL·kg·min^−1^ VO_2max_); MT = Moderate-trained (50–59 mL·kg·min^−1^ VO_2max_); HT = High-trained (60+ mL·kg·min^−1^ VO_2max_). s = significant change post-exercise (*p* < 0.05); ns = non-significant change post-exercise (*p* > 0.05); nb = no baseline resting data to compare with; c = control/placebo trial of study. # Where data have been converted from EU·mL^−1^ to pg·mL^−1^ through standard conversions (1 EU·mL^−1^ = 100 pg·mL^−1^).

**Table 5 nutrients-12-00537-t005:** Evidence basis of nutritional supplements to help protect exercise-induced GI barrier integrity loss.

Nutrient	Evidence	Dosing	Consensus and Limitations
**Carbohydrate**	Cell: − − Clinical: + + − Exercise: + + +	30–108 g·kg·h^−1^ liquid multi-transportable CHO.	Effects of pre- exercise CHO status or solid CHO ingestion unknown. Greater exploration on CHO timing and types required.
**l-Glutamine**	Cell: + + + − Clinical: + + − Exercise: + + +	0.25–0.9 g·kg·FFM.^−1^ given 1–2 h pre-exercise.	Dose ≥ 0.25 g·kg·FFM^−1^ appears favourable. High doses poorly tolerated in some individuals. No evidence during prolonged exercise or on MT.
**Bovine Colostrum**	Cell: + + + + Clinical: + + + Exercise: + +	20 g·day−1 for 14 days pre-exercise	Potentially useful following less demanding exercise. No effects with short-term supplementation. Certain formulations might be more beneficial.
**Nitric Oxide**	Cell: + + Clinical: + + Exercise: − −	More evidence required	No benefits of l-citrulline or sodium nitrate. Nitrate ingestion might compromise thermoregulation with exercise in the heat. Only two human exercise studies.
**Probiotics**	Cell: + − Clinical: + + − − Exercise: + − −	More evidence required	Contrasting results between formulations. Multi-strain probiotics seem favourable. Negative responses have been reported. Further evidence required.
**Polyphenols**	Cell: + + − − Clinical: + − Exercise: + −	3 days of 0.5 g·day^−1^ of curcumin. Quercetin not recommended	Contrasting results between formulations. Only two human exercise studies. Further evidence required.
**Zinc Carnosine**	Cell: + + + Clinical: + + Exercise: +	75 mg·day^−1^ for ≥ 2 days	Unknown effects in severe exercise situations. A 150 mg·day^−1^ dose warrants research. Only one human exercise study. Further evidence required.

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
