# Peer review of "The Gastrointestinal Exertional Heat Stroke Paradigm: Pathophysiology, Assessment, Severity, Aetiology and Nutritional Countermeasures"

_nutrients, 2020, doi:10.3390/nu12020537_

Round 1

Reviewer 1 Report

The authors have written a very nice review and I enjoyed reading it.  I only have minor comments for you to consider. The review will be a huge contribution to the literature.

There is a substantial amount of literature on farm animals about how heat stress negatively influences gut barrier function. This is especially true in poultry and pigs, but it even extends to ruminants.  Amazingly, the biology of how heat stress negatively influences GIT barrier is highly conserved amongst species (rodents, people etc..).  Arguably, the research on farm animals, heat and GIT is more advanced than in other species and this is especially obvious compared to humans. I was surprised to see that the authors either weren’t aware of this mountain of literature or chose not to incorporate it into your review.  The pig’s GIT is certainly more relevant to human medicine than is a rodents. I think heat shock proteins block NFkb activation (can’t remember the references). This likely explains why there isn’t active inflammation during EHS. The “inflammatory storm” typically happens after the heat has been removed.

Regarding the dietary interventions: here too the farm animal literature is much more advanced than we are. For example, the Zn section could be substantially improved as Zn has been a heavily studied strategy in poultry and pigs during heat stress.

Author Response

Dear Sir/Madame,

Thank you for the time and effort taken to peer review our manuscript. We are grateful that the manuscript was largely well-received. Below are our responses to address your helpful comments.

"There is a substantial amount of literature on farm animals about how heat stress negatively influences gut barrier function. This is especially true in poultry and pigs, but it even extends to ruminants.  Amazingly, the biology of how heat stress negatively influences GIT barrier is highly conserved amongst species (rodents, people etc..).  Arguably, the research on farm animals, heat and GIT is more advanced than in other species and this is especially obvious compared to humans. I was surprised to see that the authors either weren’t aware of this mountain of literature or chose not to incorporate it into your review.  The pig’s GIT is certainly more relevant to human medicine than is a rodents. I think heat shock proteins block NFkb activation (can’t remember the references). This likely explains why there isn’t active inflammation during EHS. The “inflammatory storm” typically happens after the heat has been removed."

It is assumed these comments refer to the sub-section "The GI Exertional Heat Stroke Paradigm" (Page 2-4; Lines 92-186). We agree that there is substantial evidence, across multiple species, demonstrating passive/exertional heat stress to negatively influence GI barrier integrity. However, our intention for this sub-section was only to focus on clinical literature where direct links between GI barrier integrity and CHS/EHS outcomes (e.g. mortality) were drawn. To our knowledge, this review has captured all the best available literature. We are happy to re-review this sub-section if it is apparent we have missed important evidence in livestock where direct links between GI barrier integrity loss and CHS/EHS clinical outcomes can be drawn. 

Regarding the dietary interventions: here too the farm animal literature is much more advanced than we are. For example, the Zn section could be substantially improved as Zn has been a heavily studied strategy in poultry and pigs during heat stress.

In general, the intention of this manuscript was only to include literature from studies in human subjects exposed to exertional-heat stress. To our knowledge, all research in livestock relate to prolonged passive heat stress. That said, an introductory paragraph for each nutritional countermeasure is provided, with the aim to outline key literature (in cells, animals, clinical non-heat humans) that provide a mechanistic basis for which benefits to humans during exertional-heat stress might be obtained. We agree that the literature of Zn supplementation conducted in heat-stressed livestock is strong and helps to substantiate this sub-section. In accordance, we have included the following text within this subsection (page 26; lines 789-794):

In swine, whose GI physiology closely resemble that of humans, short-term (7-17 days) supplementation (60 mg·kg·day-1) with a patented zinc amino-acid complex animal feed, reduced GI MT (endotoxin, TNF-α) during 12-14 hours of cyclic heat stress [302-303]. Similar benefits are reported on GI barrier integrity (e.g. TJ expression, villus height) following chronic zinc supplementation (~60-75 mg·kg·day-1) on other livestock (e.g. broiler chickens, bovine) when raised under cyclic heat stress [304-305].

Thank you again for your helpful comments and hope you find our responses suitable. 

Kind Regards,

Henry Ogden

Reviewer 2 Report

Well researched and comprehensive review article.

Please consider moving tables 2 to 4 to a supplement. These tables are important, however can divert the attention of the reader from important items in the text.

Author Response

Dear Sir/Madame,

Thank you for the time and effort taken to peer review our manuscript. We are grateful that the manuscript was largely well-received. 

We take your comments on-board surrounding the positioning of Tables 2-4. We were aware that the length of these tables might impact readability, but as was the importance of this literature feel it is justified to maintain the position of these tables within the main body of text.

We would be happy to accept the editorial decision surrounding the positioning of Tables 2-4 within either the main text or as a supplement. 

Kind Regards,

Henry Ogden